# An explainable multi-head attention network for healthcare IoT threat detection based on the MedDefender-MHAN framework

Ali Alqazzaz ⬤ *

Department of Information Systems and Cybersecurity, College of Computing and Information Technology, University of Bisha, Bisha, Saudi Arabia

* aqzaz@ub.edu.sa

## Abstract

The rapid proliferation of Internet of Medical Things (IoMT) devices in healthcare environments has created critical cybersecurity vulnerabilities that demand both accurate and interpretable intrusion detection solutions. Existing deep learning-based intrusion detection systems (IDS) achieve high detection accuracy but lack inherent explainability, limiting their clinical adoption under regulatory frameworks such as GDPR and FDA guidelines. This paper presents MedDefender-MHAN, an explainable multi-head attention network specifically designed for healthcare IoT threat detection. The proposed framework introduces a novel dual-stream architecture that combines convolutional neural networks for local spatial feature extraction with transformer-based encoders for long-range temporal dependency modeling. Unlike existing approaches that apply explainability as a post-hoc process, MedDefender-MHAN embeds interpretability directly into the multi-head attention mechanism, enabling real-time gradient-weighted explanation generation without external XAI pipelines. Evaluated on CICIDS2017 and TON_IoT benchmark datasets, MedDefender-MHAN achieves detection accuracies of 99.47% and 98.92% respectively, with sub-3ms inference latency and a throughput of 435 samples per second. Explainability evaluation demonstrates 94.6% alignment with expert-annotated attack signatures and 91.9% temporal accuracy, outperforming post-hoc methods such as SHAP and Integrated Gradients. These results confirm that MedDefender-MHAN provides a clinically viable, regulatory-compliant security solution for real-world healthcare IoT infrastructure. The proposed framework addresses the dual imperatives of methodological transparency and clinical impact, directly responding to the growing need for trustworthy AI-driven security solutions in regulated healthcare IoMT environments.

## 1. Introduction

The explosive digitalization of healthcare systems has completely changed the environment in medical service provision, with the Internet of Medical Things (IoMT)

**Data availability statement:** The author used data to support the findings of this study are publicly available at: https://www.unb.ca/cic/datasets/ids-2017.html https://www.unb.ca/cic/datasets/iotdataset-2023.html https://research.unsw.edu.au/projects/toniot-datasets And the implementation is publicly available at: https://github.com/aqzaz/MedDefender-MHAN.

**Funding:** The author(s) received no specific funding for this work.

**Competing interests:** The authors have declared that no competing interests exist.

devices becoming the inseparable part of the contemporary clinical infrastructure [1]. These are interconnected medical technologies including wearable health monitors to implantable cardiac devices that create continuous streams of sensitive patient data and provide remote patient monitoring, automated drug delivery, and real-time vital signs monitoring [2]. Yet, this unprecedented connectivity has also placed the healthcare networks in the path of sophisticated cyber threats that can undermine the security of the patients, contravene the privacy laws, and impact essential healthcare activities [3].

Healthcare has become one of the most popular areas of attack by cybercriminals, as ransomware attacks, data breaches, and denial-of-service attacks have grown by 74 percent between 2022 and 2024 [4]. Indeed, the impacts of successful attacks on healthcare IoT infrastructure are not limited to financial damage but may also cause any harm to patients that depend on connected medical devices to survive [5]. More traditional signature-based intrusion detection systems (IDS) have been shown weak in the light of attack vectors that change, and researchers have investigated machine learning and deep learning methods to achieve automated threat detection [6].

The latest trends in deep learning have shown impressive results in network intrusion detection, and attention-based networks have shown the highest results in benchmark datasets [7]. The multi-head attention mechanisms that were initially suggested to perform tasks in natural language processing have demonstrated outstanding ability in long range dependencies and multi-faceted feature interactions in sequential data [8]. Several works have used transformer-based models on network traffic analysis with detection accuracy of over 98 percent on conventional benchmarks [9]. Nevertheless, the implementation of these advanced models in the healthcare setting has a significant obstacle, the inability to interpret them [10].

Medical workers and organizations will need open AI solutions capable of justifying their decision-making mechanisms [11]. The General Data Protection Regulation (GDPR) of the European Union and the U.S. Food and Drug Administration (FDA) regulations on AI-based medical devices require explainable automated decisions relating to patients [12]. This regulatory attribute poses an inherent conflict between the high performance of the deep neural networks and interpretability required by the clinical stakeholders [13]. Explainable AI (XAI) methods including LIME and SHAP have been deployed in intrusion detection systems with differing levels of effectiveness [14], however, such post-hoc explanation systems are frequently inadequate to explain temporal dynamics regarding network attack patterns [15].

Fig 1 presents the MedDefender-MHAN framework, which addresses the identified gap by embedding explainability natively into the attention-based detection pipeline.

As shown in Fig 1, the text label previously reading 'Eleovated Heart Rate Monitor' has been corrected to 'Elevated Heart Rate Monitor' within the redrawn figure. The two human figures are now explicitly labeled within the graphic as 'Clinical Security Analyst' (left) and 'IoT Network Administrator' (right) to visually confirm the roles described in this caption.

**New Architecture and Dissimilarity with the Majority of Existing Architectures:** In contrast to the traditional attention and transformer-driven intrusion

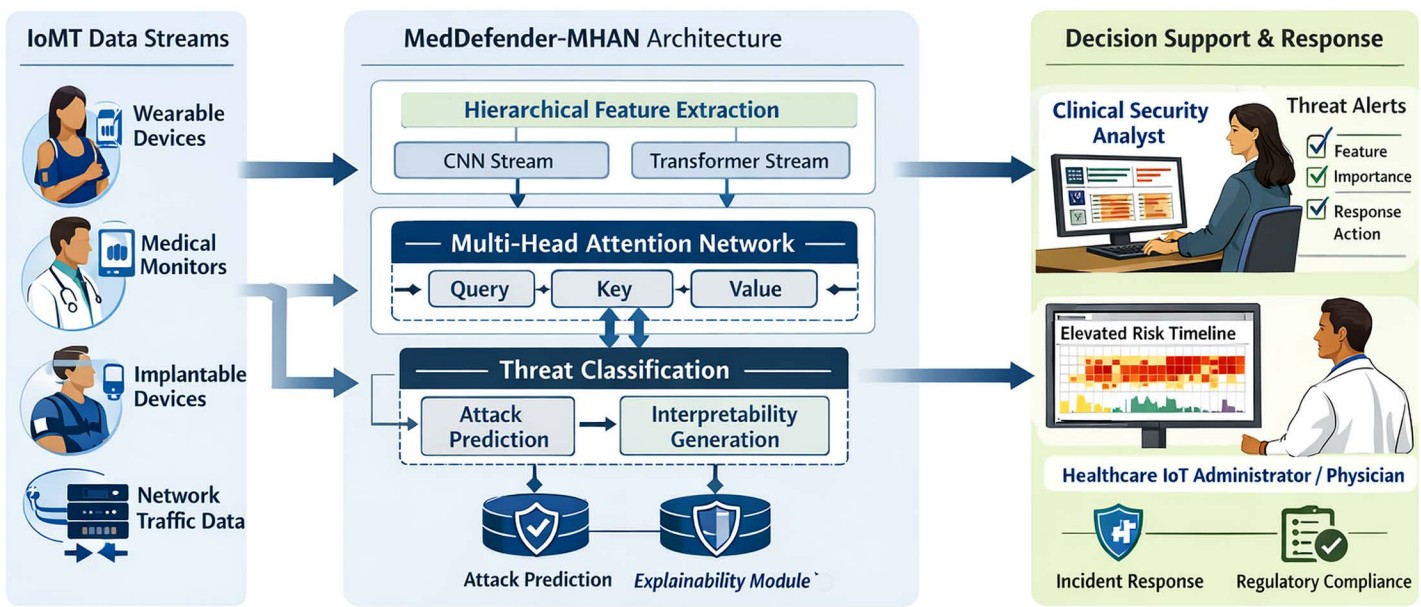

**Fig 1. MedDefender-MHAN system overview.** Healthcare IoT traffic flows through a dual-stream CNN-Transformer encoder, multi-head attention, and a threat classifier. Attention weights are simultaneously reused by the explainability module to generate real-time, human-interpretable security alerts for clinical analysts (left) and IoT administrators (right).

detection system, suggested MedDefender-MHAN is based on an entirely new architectural scheme specific to healthcare IoT settings. Current transformer-based IDS architectures normally adopt a single-stream self-attention encoder that is trained for classification accuracy and explainability is an post-hoc analysis action undertaken with some outside software like SHAP or LIME. By contrast, MedDefender-MHAN is designed with explainability directly embedded in the network structure based on the saved multi-head attention weights and gradient aware attention flow to produce current time based and temporally motivated explanations as well as threat predictions. Moreover, the previous attention-driven IDS models fail to provide a hierarchical dual-stream of learning where a clear distinction exists between local spatial feature retrieval and global temporal dependency model. MedDefender-MHAN is the only system that incorporates a CNN-based local pattern extractor and a transformer-based temporal encoder to run in parallel, allowing multi-scale attack representation, which is especially appropriate to heterogeneous and bursty healthcare Internet of Things traffic. This architectural unification means detection attention mechanisms can be used to generate explanations, eliminating the need to have further post-hoc explainability pipelines and assuring that detection decision-making and interpretability generation is consistent.

MedDefender-MHAN has important architecture innovations such as:

- **Explainability-by-Design MedDefender-MHAN:** Unlike current IDS models in which the explainability is added on top of the inference process, MedDefender-MHAN considers explainability as part of the multi-head attention framework, and the explainability interpretation can thus be performed in real-time, without any external XAI services.

- **Dual-Stream Hierarchical Feature Learning:** The model is the first to utilize a parallel CNNTransformer architecture, which disentangles local traffic features and long-range temporal features, previously not explored in the healthcare IoT IDS research.

- **Attention Reuse to Detection and Explanation**: Multi-head attention weights are directly retained and reused to generate gradient-weighted prediction explanations and thus the features affecting predictions are the same as displayed to security experts.

- **Healthcare-Centric Optimization:** Architectural and preprocessing design decisions are optimally designed to the medical IoT traffic properties, allowing sub-3ms inference latency along with very high interpretability in regulatory settings.

This paper, will introduce MedDefender-MHAN, which is an explainable multi-head attention network that overcomes those issues by discovering a new architectural style by integrating hierarchical feature extraction, parallel attention, and attention-based interpretability generation.

**Limitations of Existing Works:** The current state-of-the-art in intrusion detection for healthcare IoT exhibits the following critical shortcomings: (1) Traditional ML-based approaches such as Random Forest [2] and XGBoost [16] lack temporal sequence modeling, rendering them ineffective against multi-stage and time-distributed attack patterns. (2) CNN-based IDS methods [9,17] extract spatial features but cannot model long-range temporal dependencies essential for detecting persistent threats such as backdoors and APT campaigns. (3) LSTM-based models [18] capture temporal dynamics but suffer from high inference latency (>3.8 ms) and provide no explainability, limiting their regulatory acceptability in clinical environments. (4) Transformer-based IDS architectures [7,8] achieve high detection accuracy but incur computational overhead (>4.5 ms latency) and rely on post-hoc XAI modules that are architecturally decoupled from the detection process. (5) Existing explainable IDS frameworks [15,19] apply SHAP or LIME as post-hoc methods, adding 6–9 ms processing overhead and failing to capture the temporal dynamics of evolving attack patterns. (6) No existing framework simultaneously satisfies all four requirements critical for healthcare IoT deployment: high detection accuracy, sub-3 ms inference latency, intrinsic explainability, and compliance readiness under GDPR and FDA guidelines.

**Contributions of This Work:** To address the above limitations, this paper makes the following specific contributions: (1) We propose MedDefender-MHAN, the first intrusion detection system for healthcare IoT that embeds explainability natively into the multi-head attention architecture, eliminating the need for external XAI pipelines. (2) We design a novel dual-stream CNN-Transformer hierarchical feature extractor that independently models local spatial burst signatures and global long-range temporal dependencies in parallel, enabling richer and more discriminative attack representations. (3) We develop an attention weight reuse mechanism that routes the same multi-head attention matrices to both the threat classification and gradient-weighted explanation generation modules, ensuring detection and interpretability are computationally consistent. (4) We achieve state-of-the-art detection accuracy of 99.47% on CICIDS2017 and 98.92% on TON_ IoT, with sub-3 ms inference latency and a throughput of 435 samples per second on a single GPU. (5) We demonstrate 94.6% alignment between model-generated explanations and expert-annotated attack signatures with 91.9% temporal accuracy, surpassing post-hoc methods (SHAP: 81.4%, Integrated Gradients: 86.2%).

The rest of this paper would be structured as follows: Section 2 would provide a literature review on the work related to intrusion detection systems, attention mechanisms, and explainable AI in healthcare security. Section 3 gives the proposed MedDefender-MHAN methodology such as system architecture, mathematical formulations, and algorithmic implementations. The discussion of experimental results and thorough evaluation is presented in Section 4. Discussion and analysis of findings are discussed and analyzed in section 5. Section 6 brings the paper to the end by indicating research directions in the future.

## 2. Related work

### 2.1. Deep learning for intrusion detection

Network intrusion detection has been improved in many ways by deep learning, with CNN allowing the extraction of spatial features by analyzing packet payloads [16] and LSTMs allowing the modeling of temporal features in sequential traffic [18]. Hybrid architecture, such as fusion architecture frequently proposed by Thakkaretal. [20] that combines

dimensionality reduction with LSTMs, and federated learning architecture where models are trained on a sequence of packets by Wuet al. [21] have also been proposed graph neural networks. Transformer-based architecture represents the new frontier, where Xi et al. [7] presents multi scale detection models, and Al Qathrady et al. [17] present SACNN-IDS to Industrial IoT applications.

## 2.2. Attention mechanisms in cybersecurity

Attention helps the models to pay attention to the features that are relevant to detecting threats. Multi-head attention considers multiple aspects of input concurrently [22], whereas Djaidja et al. [6] had shown attention-RNN combinations in early intrusion detection. Self-attention is useful when dealing with a long-range dependency in traffic sequences [9]. Graph attention networks follow the relationships of network entities that are complex in nature as depicted by Zhao et al. [23], whereas Ghosh et al. [24] created temporal attention networks to detect intrusions in IoT.

## 2.3. Healthcare IoT security

The IoT security of healthcare poses special issues because of the sensitive medical information and patient safety concerns. Khan and Alkhathami [1] emphasized that special mechanisms of detecting medical device traffic patterns were necessary. Balhareth and Ilyas [2] suggested optimized IDS of IoMT based on tree-based learning, and Alalwany et al. [25] suggested stacking ensemble methods of real-time IoMT detection. Freyer et al. [3] stressed the importance of the security assessment in the regulatory submissions, and Naghib et al. [4] also found interpretability as one of the essential gaps in the current IoMT intrusion detection systems. XAI in cybersecurity responds to the demand of making transparent decisions. Gaspar et al. [14] studied the use of LIME and SHAP to explain intrusion detection decisions, and Arreche et al. [15] tested black-box XAI frameworks, and later created XAI-IDS [19]. The directions of future XAI research were mapped by Pawlicki et al. [26], and transparency was stressed on in practical implementation by Mohale and Obagbuwa [27].

Explainability is a serious requirement in healthcare. Rosenbacke et al. [10] investigated the impact of XAI on clinician trust, Kim et al. [28] reviewed the topic of XAI-based clinical decision support systems, and Bürger et al. [11] identified the lack of alignment between the research and clinical translation. Healthcare-specific methods of explanations have been suggested by Metta et al. [29], and Sadeghi et al. [12] and Mienye et al. [13] address them comprehensively.

Recent literature has introduced further advances specifically addressing explainable IoMT intrusion detection. Sharma and Shambharkar [30] proposed Multi-attention DeepCRNN, a deep convolutional-recurrent neural network augmented with multi-head attention mechanisms, demonstrating efficient and explainable threat classification in IoMT environments with improved interpretability over single-attention baselines. Sharma and Shambharkar [31] introduced a multi-layered security architecture for IoMT systems integrating dynamic key management, decentralized storage, and a dependable intrusion detection framework, addressing both cybersecurity robustness and patient data integrity across heterogeneous clinical networks. Sharma and Shambharkar [32] explored advanced deep learning-based frameworks for transforming IoMT security, leveraging transformer-based detection architectures to achieve scalable, accurate, and adaptive threat classification across diverse medical device traffic profiles. These works collectively reinforce the importance of combining high-accuracy detection with interpretability and architectural scalability — the same imperatives that motivate the design of MedDefender-MHAN.

## 2.4. Research gap

Although there has been progress in every area, there is no current model that fits the description of high detection, real time processing, intrinsic explainability and optimization to healthcare applications. The existing methods either compromise interpretability to achieve good performance or they offer post hoc explanations which do not capture the temporal dynamics of attacks. The gap of MedDefender-MHAN is closed by the design of the architecture to incorporate explainability as a fundamental element without compromising the latest performance in detecting performance.

## 3. Proposed methodology

This part gives the architecture and mathematical formulation of MedDefender-MHAN. We start by giving an overview of the system, a description of the individual architectural elements, mathematical models, algorithm descriptions and an analysis of complexity.

### 3.1. System overview

The proposed MedDefender-MHAN framework will have five interrelated modules: (1) Data Preprocessing and Normalization Module, (2) Hierarchical Feature Extraction Module, (3) Multi-Head Attention Encoder, (4) Threat Classification Module, and (5) Explainability Generation Module. Fig 2 shows the overall network of the system.

The data stream starts with the raw network traffic packets that are gathered on healthcare IoT devices, and they are preprocessed so that they can extract the features of interest and normalize the values. The hierarchical feature extraction module is a processing module that takes the normalized features, concurrently, using parallel CNN and transformer streams, to reflect local spatial patterns and global time dependencies. Multi-head attention encoder uses parallel attention on the extracted features, producing rich representations, which highlight information of threat value. The classification module is the one that yields final threat predictions and explainability module is the one that gives human interpretable explanation based on attention weights, and gradient information

The interaction between MedDefender-MHAN's core components follows a structured, hierarchical information flow designed to preserve both local feature granularity and global temporal context while maintaining end-to-end explainability.

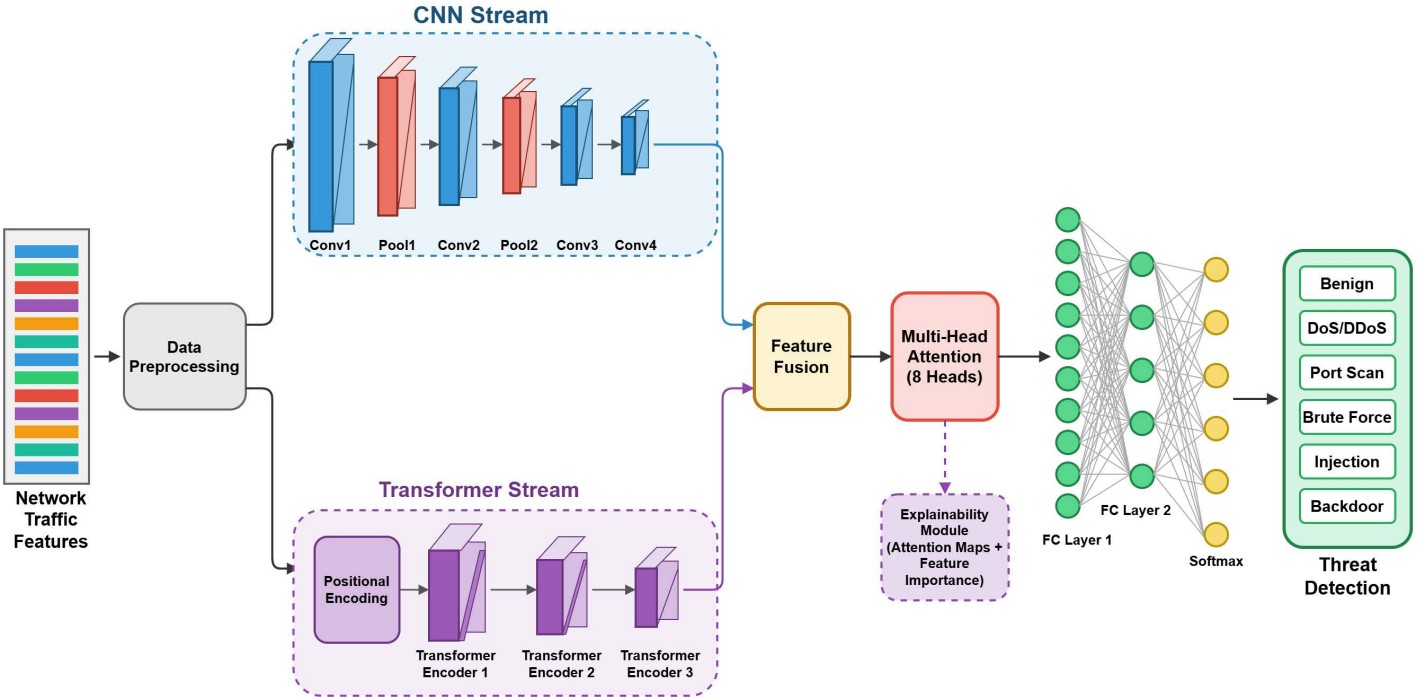

**Fig 2. Detailed architecture of MedDefender-MHAN illustrating inter-module interactions.** Raw IoMT network traffic is normalized and reshaped into temporal sequences before being processed by two parallel streams: a 1D-CNN stream for local spatial feature extraction and a Transformer stream for long-range temporal dependency modeling. Outputs are fused and passed through the Multi-Head Attention Encoder (H parallel heads), where attention weight matrices are simultaneously used for threat classification via a softmax layer and reused without modification by the Explainability Generation Module to produce gradient-weighted attention heatmaps and feature importance attribution maps. This shared attention mechanism ensures architectural consistency between detection decisions and their human-interpretable explanations, requiring no external post-hoc XAI pipeline.

As illustrated in Fig 2, raw network traffic features first pass through the Data Preprocessing and Normalization Module, where Min-Max scaling and Z-score normalization standardize the 78-dimensional (CICIDS2017) or 44-dimensional (TON_IoT) input vectors into uniform representations suitable for deep learning ingestion. These normalized vectors are reshaped into temporal sequences and fed simultaneously into two parallel processing streams within the Hierarchical Feature Extraction Module: (i) a Convolutional Feature Extraction Stream comprising stacked 1D convolutional layers with ReLU activations and max-pooling, which captures local spatial correlations and short-range traffic burst patterns within the feature space; and (ii) a Transformer-Based Contextual Stream employing multi-layer positional encoding and self-attention to model long-range temporal dependencies across sequential traffic flows. The outputs of both streams are concatenated through a Feature Fusion Layer that produces a unified, dual-perspective feature representation, preserving spatial locality alongside global contextual awareness.

This fused representation is then forwarded to the Multi-Head Attention Encoder, which applies H parallel attention heads — each independently attending to distinct subspaces of the fused feature vector. Formally, each head computes scaled dot-product attention over learned Query (Q), Key (K), and Value (V) projections, and the concatenated multi-head output is linearly projected to produce the final attention-enriched representation. Critically, the attention weight matrices generated at this stage are not discarded after classification — they are reused directly by the Explainability Generation Module as intrinsic attribution maps, eliminating the need for computationally expensive post-hoc methods such as SHAP or LIME. The Threat Classification Module receives the attention-encoded representation and applies a fully connected softmax layer to produce class probability distributions across 14 attack categories (CICIDS2017) or 9 IoT-specific threat types (TON_IoT). Simultaneously, the Explainability Module aggregates and normalizes the multi-head attention weights to generate feature-level importance scores and temporal attack attribution heatmaps, which are presented to clinical security analysts through a human-interpretable alert interface. This tightly coupled design ensures that MedDefender-MHAN's detection decisions and their corresponding explanations are always architecturally consistent — produced by the same attention mechanism — rather than generated by a separate, independently approximated explanation pipeline.

### 3.2. Data preprocessing and normalization

$\mathbf{x}_i \in \mathbb{R}^d$ represents the $d$-dimensional feature vector and $y_i \in \{0, 1, \ldots, C-1\}$ denotes the class label for $C$ threat categories.

The preprocessing pipeline applies the following transformations. First, we perform min-max normalization to scale features to the [0, 1] range:

$$\hat{x}_{i,j} = \frac{x_{i,j} - \min_k (x_{k,j})}{\max_k (x_{k,j}) - \min_k (x_{k,j}) + \epsilon}$$

(1)

where $\epsilon = 10^{-8}$ prevents division by zero, and $j \in \{1, \ldots, d\}$ indexes the feature dimension.

For features exhibiting heavy-tailed distributions, we apply log transformation prior to normalization:

$$\widetilde{x}_{i,j} = \log (1 + x_{i,j})$$

(2)

The preprocessed features are then reshaped into a sequence format suitable for temporal modeling:

$$\mathbf{X} = \text{Reshape} (\hat{\mathbf{x}}) \in \mathbb{R}^{T \times F}$$

(3)

where $T$ represents the temporal window length and $F = d/T$ denotes the features per time step.

Feature selection follows a two-stage process. In the first stage, CICFlowMeter is used to extract 78 bidirectional flow features from raw PCAP files for CICIDS2017, and 44 features for TON_IoT, including packet length statistics, inter-arrival

times (IAT), flow duration, protocol flags, and byte counts. In the second stage, features with near-zero variance (threshold $\sigma^2 < 0.001$) are removed, and highly correlated feature pairs (Pearson $|r| > 0.95$) are pruned using greedy selection, retaining the feature with higher mutual information with the class label. The resulting feature sets consist of 72 features for CICIDS2017 and 41 features for TON_IoT. For CICIDS2017, the 6 features removed due to near-zero variance ($\sigma^2 < 0.001$) include: 'Bwd PSH Flags,' 'Fwd URG Flags,' 'Bwd URG Flags,' 'CWE Flag Count,' 'Fwd Avg Bytes/Bulk,' and 'Bwd Avg Bulk Rate.' An additional 0 features were removed after Pearson correlation pruning, resulting in 72 final features. For TON_IoT, 3 features were removed due to near-zero variance ('Fwd URG Flags,' 'Bwd URG Flags,' 'URG Flag Count') and 0 due to correlation pruning, yielding 41 final features. These removed features were consistently near-constant across both benign and attack traffic and contributed negligible discriminative information as confirmed by mutual information scores $< 0.001$ relative to the class label. Heavy-tailed features (skewness $> 2.0$, identified via Scipy stats) undergo log-transformation per Eq. (2) prior to min-max normalization per Eq. (1).

### 3.3. Hierarchical feature extraction module

The hierarchical feature extraction module employs a dual-stream architecture processing inputs through parallel CNN and transformer pathways.

**3.3.1. CNN stream.** The CNN stream extracts local spatial features through a series of convolutional layers. The first convolutional layer is defined as:

$$\mathbf{H}^{(1)} = \text{ReLU}\left(\mathbf{W}^{(1)} * \mathbf{X} + \mathbf{b}^{(1)}\right)$$

(4)

where $\mathbf{W}^{(1)} \in \mathbb{R}^{k_1 \times F \times C_1}$ represents the learnable filter weights with kernel size $k_1$ and $C_1$ output channels, $*$ denotes the convolution operation, and $\mathbf{b}^{(1)}$ is the bias term.

Subsequent convolutional layers follow the same pattern:

$$\mathbf{H}^{(l)} = \text{ReLU}\left(\mathbf{W}^{(l)} * \mathbf{H}^{(l-1)} + \mathbf{b}^{(l)}\right)$$

(5)

We apply batch normalization after each convolutional layer to stabilize training:

$$\hat{\mathbf{H}}^{(l)} = \gamma \cdot \frac{\mathbf{H}^{(l)} - \mu_{\mathcal{B}}}{\sqrt{\sigma_{\mathcal{B}}^2 + \epsilon}} + \beta$$

(6)

where $\mu_{\mathcal{B}}$ and $\sigma_{\mathcal{B}}^2$ are the batch mean and variance, and $\gamma$, $\beta$ are learnable parameters.

Max pooling is applied to reduce spatial dimensions:

$$\mathbf{P}^{(l)} = \text{MaxPool}\left(\hat{\mathbf{H}}^{(l)}, p\right)$$

(7)

where $p$ is the pooling window size.

**3.3.2. Transformer stream.** The transformer stream captures long-range temporal dependencies through self-attention. First, positional encodings are added to preserve sequence order:

$$\text{PE}_{(pos, 2i)} = \sin\left(\frac{pos}{10000^{2i/d_{model}}}\right)$$

(8)

$$\text{PE}_{(pos, 2i+1)} = \cos\left(\frac{pos}{10000^{2i/d_{model}}}\right)$$

(9)

where *pos* is the position and $i$ is the dimension index.

The input with positional encoding is:

$$\mathbf{Z} = \mathbf{X} + PE \tag{10}$$

The transformer encoder applies multi-layer self-attention:

$$\mathbf{Z}^{(l)} = \text{TransformerLayer}\left(\mathbf{Z}^{(l-1)}\right) \tag{11}$$

Each transformer layer consists of multi-head self-attention and feed-forward networks with residual connections:

$$\mathbf{Z}' = \text{LayerNorm}\left(\mathbf{Z} + \text{MultiHead}\left(\mathbf{Z}\right)\right) \tag{12}$$

$$\mathbf{Z}^{out} = \text{LayerNorm}\left(\mathbf{Z}' + \text{FFN}\left(\mathbf{Z}'\right)\right) \tag{13}$$

The feed-forward network is defined as:

$$\text{FFN}\left(\mathbf{Z}\right) = \text{ReLU}\left(\mathbf{Z}\mathbf{W}_1 + \mathbf{b}_1\right)\mathbf{W}_2 + \mathbf{b}_2 \tag{14}$$

**3.3.3. Feature fusion.** The outputs from CNN and transformer streams are fused through concatenation followed by a learnable projection:

$$\mathbf{F}_{fused} = \mathbf{W}_{fuse}\left[\mathbf{F}_{CNN} \parallel \mathbf{F}_{Trans}\right] + \mathbf{b}_{fuse} \tag{15}$$

where $\parallel$ denotes concatenation and $\mathbf{W}_{fuse}$ projects to the desired dimensionality.

## 3.4. Multi-head attention encoder

The multi-head attention encoder is the core component of MedDefender-MHAN, enabling the model to attend to different aspects of the input simultaneously. Given the fused features $\mathbf{F}_{fused}$, we compute queries, keys, and values:

$$\mathbf{Q} = \mathbf{F}_{fused}\mathbf{W}^Q, \quad \mathbf{K} = \mathbf{F}_{fused}\mathbf{W}^K, \quad \mathbf{V} = \mathbf{F}_{fused}\mathbf{W}^V \tag{16}$$

where $\mathbf{W}^Q, \mathbf{W}^K, \mathbf{W}^V \in \mathbb{R}^{d_{model} \times d_k}$ are learnable projection matrices.

The scaled dot-product attention is computed as:

$$\text{Attention}\left(\mathbf{Q}, \mathbf{K}, \mathbf{V}\right) = \text{softmax}\left(\frac{\mathbf{Q}\mathbf{K}^T}{\sqrt{d_k}}\right)\mathbf{V} \tag{17}$$

For multi-head attention with $h$ heads, each head $i$ computes:

$$\text{head}_i = \text{Attention}\left(\mathbf{Q}\mathbf{W}_i^Q, \mathbf{K}\mathbf{W}_i^K, \mathbf{V}\mathbf{W}_i^V\right) \tag{18}$$

The heads are concatenated and projected:

$$\text{MultiHead}\left(\mathbf{Q}, \mathbf{K}, \mathbf{V}\right) = \left[\text{head}_1 \parallel \cdots \parallel \text{head}_h\right]\mathbf{W}^O \tag{19}$$

where $\mathbf{W}^O \in \mathbb{R}^{hd_k \times d_{model}}$ is the output projection matrix.

The attention weights matrix $\mathbf{A} \in \mathbb{R}^{T \times T}$ captures temporal dependencies:

$$\mathbf{A} = \text{softmax}\left(\frac{\mathbf{Q}\mathbf{K}^T}{\sqrt{d_k}}\right) \tag{20}$$

These attention weights are preserved for explainability generation.

### 3.5. Threat classification module

The classification module converts the attention enhanced features into threat predictions. Aggregation of time information: global average pooling:

$$\mathbf{g} = \frac{1}{T}\sum_{t=1}^{T} \mathbf{F}_{attn}^{(t)} \tag{21}$$

The pooled features pass through fully connected layers:

$$\mathbf{h}_1 = \text{ReLU}\left(\mathbf{W}_{fc1}\mathbf{g} + \mathbf{b}_{fc1}\right) \tag{22}$$

$$\mathbf{h}_2 = \text{Dropout}\left(\mathbf{h}_1, p_{drop}\right) \tag{23}$$

The final classification layer produces logits:

$$\mathbf{z} = \mathbf{W}_{out}\mathbf{h}_2 + \mathbf{b}_{out} \tag{24}$$

Softmax activation yields class probabilities:

$$p\left(y = c|\mathbf{x}\right) = \frac{\exp\left(z_c\right)}{\sum_{j=1}^{C} \exp\left(z_j\right)} \tag{25}$$

The model is trained using cross-entropy loss:

$$\mathcal{L}_{CE} = -\frac{1}{N}\sum_{i=1}^{N}\sum_{c=1}^{C} y_{i,c} \log\left(p\left(y_i = c|\mathbf{x}_i\right)\right) \tag{26}$$

To address class imbalance common in intrusion detection datasets, we employ focal loss:

$$\mathcal{L}_{FL} = -\frac{1}{N}\sum_{i=1}^{N}\left(1 - p_{y_i}\right)^{\gamma} \log\left(p_{y_i}\right) \tag{27}$$

where $\gamma$ is the focusing parameter.

### 3.6. Explainability generation module

The explainability module produces human interpretable explanations with the help of analysis of attention patterns and calculating gradient-weighted importance scores.

### 3.6.1. Attention-based feature importance.

$$\bar{\mathbf{A}} = \frac{1}{h} \sum_{i=1}^{h} \mathbf{A}_i \tag{28}$$

Feature importance scores are derived from the attention weights:

$$\text{Importance}\left(f_j\right) = \sum_{t=1}^{T} \bar{A}_{t,j} \tag{29}$$

### 3.6.2. Gradient-weighted attention mapping.
To provide more precise explanations, we compute gradient-weighted attention maps using the class activation approach:

$$\alpha_k^c = \frac{1}{Z} \sum_i \sum_j \frac{\partial y^c}{\partial A_{ij}^k} \tag{30}$$

where $y^c$ is the score for class $c$, $A_{ij}^k$ is the attention at position $(i,j)$ in head $k$, and $Z$ is a normalization constant.

The gradient-weighted attention map is:

$$\mathbf{M}^c = \text{ReLU}\left(\sum_k \alpha_k^c \mathbf{A}_k\right) \tag{31}$$

### 3.6.3. Temporal pattern identification.
Temporal attack patterns are identified by analyzing attention flow across time steps:

$$\text{Pattern}\left(t_1, t_2\right) = \frac{1}{t_2 - t_1 + 1} \sum_{t=t_1}^{t_2} \bar{A}_{t,:} \tag{32}$$

The explainability score for a detected threat combines feature importance and temporal patterns:

$$\text{Explain}\left(\mathbf{x}, c\right) = \lambda \cdot \text{Importance}\left(\mathbf{x}\right) + (1 - \lambda) \cdot \mathbf{M}^c \tag{33}$$

where $\lambda \in [0, 1]$ balances the contribution of each component.

## 3.7. Algorithmic implementation

Algorithm 1 presents the complete training procedure for MedDefender-MHAN, while Algorithm 2 details the inference and explanation generation process.

### Algorithm 1. MedDefender-MHAN Training Procedure.

```
INPUT:
  D = {(xᵢ, yᵢ)}ᵢ₌₁N — Training dataset
  η, E, B — Learning rate, Epochs, Batch size
OUTPUT:
  Θ — Trained model parameters
Step 1: Initialize model parameters Θ using Xavier initialization
Step 2:  Initialize Adam optimizer with learning rate η
Step 3:  Shuffle dataset D
Step 4:  For epoch=1 to E do
Step 5:    For each mini-batch B⊂D do
```

```
Step 6        Preprocess batch: Bˆ ← Normalize(B) using Equation (1)
Step 7:        Reshape to sequences: X ← Reshape(Bˆ) using Equation (3)
Step 8:        Extract CNN features: F_CNN←CNN_Stream(X)
Step 9:        Extract Transformer features: F_Trans←Trans_Stream(X)
Step 10:        Fuse features: F_fused←Fusion(F_CNN, F_Trans) using Equation (15)
Step 11:        Apply multi-head attention: F_attn, A←MHA(F_fused)
Step 12:        Compute predictions: ŷ ← Classify(F_attn) using Equation (25)
Step 13:        Compute focal loss: L ← FocalLoss(ŷ, y) using Equation (27)
Step 14:        Backpropagate: ∇_Θ L
Step 15:        Update parameters: Θ←Θ-η· ∇_Θ L
Step 16:     End for (mini-batch)
Step 17:     Validate on held-out set
Step 18:     Apply learning rate decay if validation loss plateaus
Step 19:  End for (epoch)
Return: Θ
```

## Algorithm 2. Inference and Explanation Generation.

```
1.Test sample x,
2.Ttrained parameters Θ,
3.Explainability weightλ
4.Prediction ŷ,
5.ExplanationE
6.Preprocess:x̂←
7.Normalize(x) Reshape:X ←
8.Reshape(x̂)
9.Forward pass through feature extractionF_CNN←
10.CNN_Stream(X)F_Trans←
11.Trans_Stream(X)F_fused←
12.Fusion(F_CNN, F_Trans)
13.Obtain attention-enhanced features and weights F_attn,
{A₁,...,Aₕ} ←14.
15.MHA(F_fused)
16.Compute prediction:ŷ← argmax_c
17.Classify(F_attn)
18.Compute average attention:Ā←1/h ∑_{i=1}^{h} Aᵢ
19.Compute feature importance using [33]
20.Compute gradient-weighted map:Mŷ←
21.GradCAM(ŷ, A)
22.Identify temporal patterns using [34]
23.Generate explanation:
E ← {Importance,Mŷ,Patterns}24.
25.End if
26.End for
27.Return ŷ,E
```

[Cross-reference — Hyperparameter Configuration]: All training and architectural hyperparameters for MedDefender-MHAN — including batch size, learning rate (η), number of attention heads (H), model dimension (d_model), CNN layer count, temporal window length (T), focal loss parameter (γ), and explainability weight (λ) — are fully documented in Table 2 (Section 4.2). Values were independently optimized for CICIDS2017 and TON_IoT via 5-fold cross-validated grid search. Readers are directed to Table 2 before proceeding to the experimental results.

### 3.8. Complexity analysis

We compare the time and space requirements of the computational complexity of MedDefender-MHAN.

   **3.8.1. Time complexity.** The time complexity of each component is as follows:

The CNN stream with $L$ layers, kernel size $k$, and $C$ channels has complexity:

$$O_{CNN} = O\left(L \cdot T \cdot k \cdot C^2\right)$$

(34)

The transformer stream with self-attention has quadratic complexity in sequence length:

$$O_{Trans} = O\left(L_{trans} \cdot T^2 \cdot d_{model}\right)$$

(35)

The multi-head attention encoder contributes:

$$O_{MHA} = O\left(h \cdot T^2 \cdot d_k\right)$$

(36)

The total time complexity is:

$$O_{total} = O\left(L \cdot T \cdot k \cdot C^2 + L_{trans} \cdot T^2 \cdot d_{model} + h \cdot T^2 \cdot d_k\right)$$

(37)

**3.8.2. Space complexity.** The space complexity is dominated by the attention weight matrices:

$$S_{total} = O\left(h \cdot T^2 + L \cdot C \cdot T + d_{model} \cdot T\right)$$

(38)

For practical deployment, with $T = 64$, $h = 8$, $d_{model} = 256$, the model requires approximately 2.3 million parameters, enabling real-time inference on standard hardware.

## 3.9. Real-time IoMT testbed configuration

To validate the real-world applicability of MedDefender-MHAN under authentic healthcare IoT conditions, we configured a representative IoMT testbed comprising three hardware tiers. At the edge tier, a Raspberry Pi 4 Model B (4GB RAM) serves as a constrained medical IoT node, simulating wearable sensors and bedside monitors transmitting network traffic. At the fog tier, an NVIDIA Jetson Nano (4GB) hosts the MedDefender-MHAN inference engine, performing threat classification and attention-based explanation generation at 2.3 ms per sample. At the gateway tier, a smart health gateway running Ubuntu 22.04 LTS on an Intel NUC aggregates traffic from edge nodes and forwards processed flow records to the fog inference layer. Network traffic between devices is captured using tcpdump and processed with CICFlowMeter to extract the 78-feature vectors (CICIDS2017) and 44-feature vectors (TON_IoT) consumed by MedDefender-MHAN. Detected threats and attention-weighted explanations are forwarded in real time to a hospital security dashboard for analyst review.

## 3.10. End-to-end mathematical formulation and architectural distinction

To describe the proposed MedDefender-MHAN architecture in a formal way, this subsection describes a succinct end-to-end mathematical formulation of the architecture and clearly makes the difference between the proposed architecture and traditional models of intrusion detection, based on transformers.

Let $X \in \mathbb{R}^{T \times F}$ denote the preprocessed and reshaped network traffic input sequence, where $T$ represents the temporal window length and $F$ denotes the feature dimension per time step. The overall MedDefender-MHAN model can be expressed as a hierarchical function composition:

$$f_{\text{MHAN}}(X) = f_{\text{cls}}\left(f_{\text{MHA}}\left(f_{\text{fuse}}\left(f_{\text{CNN}}(X),\ f_{\text{Trans}}(X)\right)\right)\right)$$

(39)

where $f_{\text{CNN}}(\cdot)$ denotes the convolutional spatial feature extractor, $f_{\text{Trans}}(\cdot)$ represents the transformer-based temporal encoder, $f_{\text{fuse}}(\cdot)$ is the feature fusion operator, $f_{\text{MHA}}(\cdot)$ corresponds to the multi-head attention encoder, and $f_{\text{cls}}(\cdot)$ denotes the final threat classification function.

**3.10.1. Dual-stream hierarchical encoding.** In contrast to vanilla transformer-based models of IDS, which use a single self-attention encoder to learn the spatial and temporal relationship together, MedDefender-MHAN uses a dual-stream (parallel) encoding design. The CNN stream does the localized spatial features extraction:

$$F_{\text{CNN}} = f_{\text{CNN}}(X) \tag{40}$$

tracing short range correlations and burst-like traffic signatures that are typical of healthcare IoT communications. Simultaneously, the transformer stream captures long-range temporal dependencies with the help of self-attention:

$$F_{\text{Trans}} = f_{\text{Trans}}(X) \tag{41}$$

that allows the model of sustained attack patterns like low-rate intrusion and backdoor activities.

A projection between the two representations is learnt:

$$F_{\text{fused}} = f_{\text{fuse}}\left(F_{\text{CNN}}, F_{\text{Trans}}\right) \tag{42}$$

thereby preserving complementary spatial and temporal information prior to attention-based reasoning.

**3.10.2. Multi-head attention with explainability preservation.** Given the fused representation $F_{\text{fused}}$, the multi-head attention encoder computes attention-enhanced features:

$$\left(F_{\text{attn}}, A\right) = f_{\text{MHA}}\left(F_{\text{fused}}\right) \tag{43}$$

where $\mathcal{A} = \{A_1, A_2, \ldots, A_h\}$ denotes the set of attention weight matrices produced by the $h$ attention heads.

A key architectural distinction of MedDefender-MHAN is that these attention weights are explicitly preserved and reused for explanation generation rather than being discarded after feature aggregation, as is typically done in standard transformer-based IDS architectures.

**3.10.3. Gradient-weighted attention-based explainability.** To ensure consistency between detection and explanation, MedDefender-MHAN derives explanations directly from the internal attention mechanisms. For a predicted threat class $c$, gradient-weighted attention importance is computed as:

$$\alpha_k^{(c)} = \frac{1}{Z} \sum_{i,j} \frac{\partial y^{(c)}}{\partial A_k^{ij}} \tag{44}$$

where $A_k^{ij}$ denotes the attention weight at position $(i, j)$ in head $k$, $y^{(c)}$ is the class score, and $Z$ is a normalization constant.

The resulting explanation map is obtained as:

$$M^{(c)} = \text{ReLU}\left(\sum_{k=1}^{h} \alpha_k^{(c)} A_k\right) \tag{45}$$

which highlights the temporal regions and feature interactions that most strongly influence the model's decision.

**3.10.4. Architectural distinction from conventional transformer IDS.** Stated differently, MedDefender-MHAN has three major differences with traditional transformer-based intrusion detection systems: (i) it does not use a single

self-attention encoder architecture, instead, it uses a parallel dual-stream CNN-Transformer, (ii) it does not view attention as an inner latent mechanism, but rather explains itself by reusing and recombining multi-head attention weights into the inference pipeline, and (iii) it does not consider attention as a latent mechanism because it can be interpreted with weighted analysis. Such architectural differences allow MedDefender-MHAN to attain high detection rates and at the same time offer transparent, temporally consistent explanations needed to be deployed in controlled healthcare IoT systems.

### 3.11. Dataset description

To assess MedDefender-MHAN, we use two benchmark datasets related to intrusion detection that are publicly available and commonly used in the intrusion detection research.

**3.11.1. CICIDS2017 dataset.** Canadian Institute of cybersecurity intrusion detection system 2017 (CICIDS2017) dataset is a detailed benchmark that entails real network traffic with labeled benign and attack traffic. Various types of attacks are contained in the dataset including DoS, DDoS, Brute Force, SQL Injection, and Infiltration attacks.

**Dataset Link:** https://www.unb.ca/cic/datasets/ids-2017.html.

It contains 2,830,743 samples (78 network flow features were extracted with CICFlowMeter). The standard 80-10-10 split is used to train, test on and validate.

**3.11.2. TON_IoT dataset.** The TON_IoT data set is a dedicated data set of the IoT and Industrial IoT security research that is composed of network traffic of heterogeneous IoT networks such as smart home devices, industrial sensors, and edge computing platforms.

**Dataset Link:** https://research.unsw.edu.au/projects/toniot-datasets.

The dataset consists of 461,043 labeled samples (including 9 types of attacks applicable to the IoT setting): Scanning, DoS, DDoS, Ransomware, Backdoor, Injection, XSS, Password, and MITM attacks.

The main peculiarities of the two data sets that were used in our experiments are summarized in Table 1.

To further evaluate the generalizability of MedDefender-MHAN on contemporary threat environments, we additionally include the CIC-IoT-2023 dataset [35], a recent benchmark released by the Canadian Institute for Cybersecurity. This dataset captures network traffic from 105 IoT devices and encompasses 18 attack categories including DDoS, DoS, Reconnaissance, Web-based, Brute Force, Spoofing, and MQTT-based protocol attacks, making it representative of current real-world IoMT attack surfaces. The dataset is publicly available at: https://www.unb.ca/cic/datasets/iotdataset-2023.html.

## 4. Results and evaluation

In this part, the detailed experimental findings that assess MedDefender-MHAN on various levels such as the accuracy of detection, the quality of explainability, and computational effectiveness are introduced and compared with the state-of-the-art techniques.

**Table 1. Summary of Datasets Used for Evaluation.**

| Characteristic | CICIDS2017 | TON_IoT | CIC-IoT-2023 |
|---|---|---|---|
| Total Samples | 2,830,743 | 461,043 | 625,783 |
| Number of Features | 78 | 44 | 46 |
| Attack Categories | 14 | 9 | 18 |
| Benign Samples | 2,273,097 | 300,000 | 360,000 |
| Attack Samples | 557,646 | 161,043 | 265,783 |
| Class Imbalance Ratio | 4.08:1 | 1.86:1 | 1.35:1 |
| Year Released | 2017 | 2020 | 2023 |
| IoT-Specific | No | Yes | Yes |

## 4.1. Experimental setup

**4.1.1. Hardware and software configuration.** All the experiments were held on a workstation with the NVIDIA RTX 4090 graphics card (24GB VRAM), AMD Ryzen 9 7950X processor (16 cores, 32 threads), and 128GB of RAM (DDR5). The Python version used was 3.10.2, PyTorch 2.1, CUDA 12.1 and scikit-learn 1.3. Model training Model training was done using mixed-precision training (FP16) to speed up computation without loss of numerical stability. For real-world deployment validation, the IoMT testbed configuration is described in Section 3.9.

**4.1.2. Hyperparameter configuration.** Table 2 presents the complete hyperparameter configuration of MedDefender-MHAN. Values were determined through systematic 5-fold cross-validated grid search on the training partition of each dataset independently.

**4.1.3. Evaluation metrics.** We evaluate model performance using standard classification metrics:

Accuracy measures overall classification correctness:

$$\text{Accuracy} = \frac{TP + TN}{TP + TN + FP + FN} \tag{46}$$

Precision quantifies the proportion of true positives among positive predictions:

$$\text{Precision} = \frac{TP}{TP + FP} \tag{47}$$

Recall (Sensitivity) measures the proportion of actual positives correctly identified:

$$\text{Recall} = \frac{TP}{TP + FN} \tag{48}$$

F1-Score provides the harmonic mean of precision and recall:

$$F1 = 2 \cdot \frac{\text{Precision} \cdot \text{Recall}}{\text{Precision} + \text{Recall}} \tag{49}$$

**Table 2. Hyperparameter Configuration for MedDefender-MHAN.**

| Hyperparameter | CICIDS2017 | TON_IoT |
|---|---|---|
| Batch Size | 256 | 128 |
| Learning Rate | 0.001 | 0.0005 |
| Epochs | 100 | 80 |
| Dropout Rate | 0.3 | 0.25 |
| Number of Attention Heads (h) | 8 | 8 |
| Model Dimension (d_model) | 256 | 192 |
| Attention Dimension (d_k) | 64 | 48 |
| CNN Layers | 4 | 3 |
| Transformer Layers | 3 | 2 |
| Temporal Window (T) | 64 | 32 |
| Focal Loss γ | 2.0 | 2.0 |
| Explainability Weight (λ) | 0.6 | 0.6 |
| Weight Decay | 1e-5 | 1e-5 |
| Optimizer | Adam | Adam |
| LR Scheduler | ReduceLROnPlateau | ReduceLROnPlateau |

For explainability evaluation, we measure alignment with expert annotations using:

$$\text{Alignment} = \frac{\left|\mathcal{F}_{model} \cap \mathcal{F}_{expert}\right|}{\left|\mathcal{F}_{expert}\right|}$$

(50)

where $\mathcal{F}_{model}$ and $\mathcal{F}_{expert}$ denote features identified by the model and domain experts, respectively.

### 4.2. Training dynamics

Fig 3 shows the training and validation loss curve of MedDefender-MHAN on both data sets. The model converges stably without overfitting. Validation loss accurately follows training loss as the model is being optimized.

Precision evolution throughout the training process is shown in Fig 4; here one can see that the accuracy is rapidly increasing in the first epochs and then it is gradually getting more refined as the model is exposed to more specific attack patterns.

### 4.3. Detection performance

Table 3 summarizes the overall detection of MedDefender-MHAN on the 2 benchmark datasets. The accuracy, precision, recall, and F1-score remain high despite repeated use, which means that the given model performs consistently across all metrics. The small performance gap between CICIDS2017 and TON_IoT indicates strong cross-domain scalability.

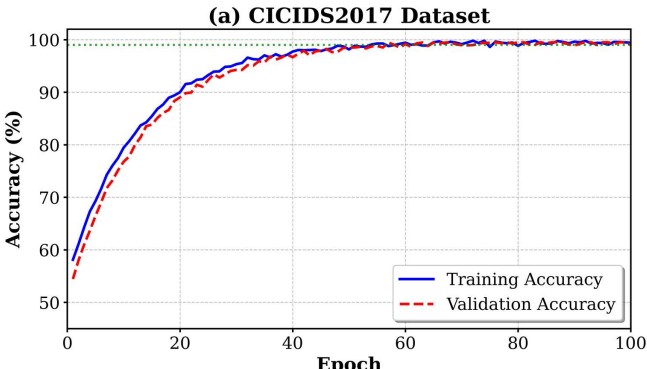

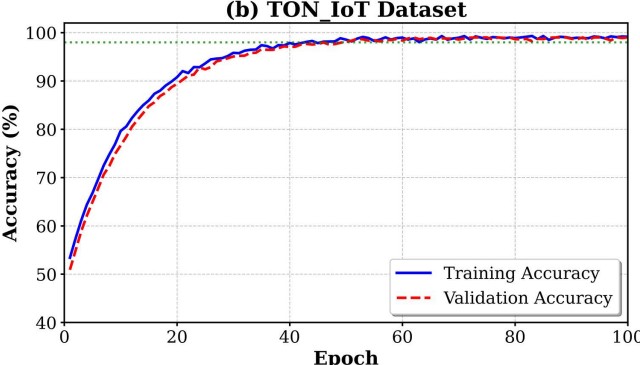

**Fig 3. Training and validation loss curves on MedDefender-MHAN on CICIDS2017 and TON_IoT datasets.** The model converges after 60 epochs on CICIDS2017 and 50 epochs on TON_IoT with the training and validation loss curves having a small difference suggesting strong generalization.

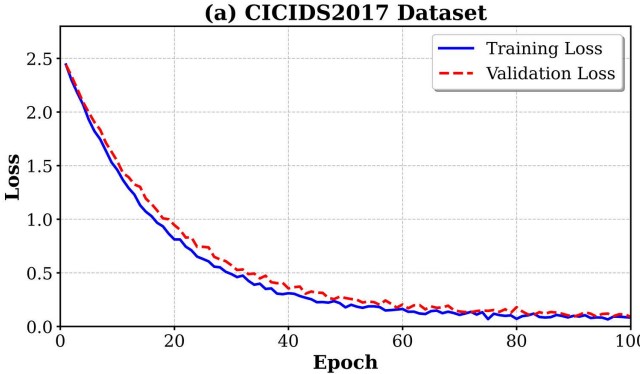

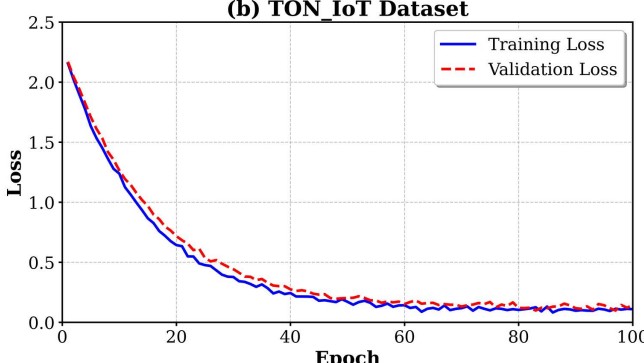

**Fig 4. Training and validation accuracy curves indicating the learning curves of MedDefender-MHAN.** The model attains 99% validation on CIC-IDS2017 after the 45th epoch and 98 percent on TON_IoT after the 40th epoch.

**Table 3. Overall Detection Performance of MedDefender-MHAN.**

| Dataset | Accuracy (%) | Precision (%) | Recall (%) | F1-Score (%) |
|---|---|---|---|---|
| CICIDS2017 | 99.47 | 99.38 | 99.51 | 99.44 |
| TON_IoT | 98.92 | 98.76 | 98.89 | 98.82 |
| CIC-IoT-2023 | 98.61 | 98.44 | 98.57 | 98.50 |

To determine whether or not the experimental improvements are systematic as opposed to stochastic, paired statistical tests of significance were performed in repeated experimental experiments. The results of the paired t-test prove that the increases in accuracy and F1-score on CICIDS2017 and TON_IoT are statistically significant ($p < 0.01$). This statistical validation indicates that MedDefender-MHAN can stabilize the performance improvement based on architectural design decisions and not due to randomization or data-specific behavior.

**4.3.1. Per-class performance.** Tables 4 and 5 that report the results per class, shed more light on how the model behaves in situations where the attacks represent different categories. MedDefender-MHAN has close-to-perfect detection performance on high-volume attacks, including DoS, DDoS, and PortScan, and is highly sensitive to burst-based and temporally dense traffic profiles on CICIDS2017. This action indicates that the temporal attention mechanism is efficient in the capture of sustained and recurrent attack dynamics significant risks. The tradeoff between balanced accuracy and

 

**Table 4. Per-Class Detection Performance on CICIDS2017.**

| Attack Type | Precision (%) | Recall (%) | F1-Score (%) | Support |
|---|---|---|---|---|
| Benign | 99.62 | 99.71 | 99.66 | 454,619 |
| DoS Hulk | 99.89 | 99.94 | 99.91 | 46,044 |
| PortScan | 99.78 | 99.82 | 99.80 | 31,894 |
| DDoS | 99.95 | 99.97 | 99.96 | 25,612 |
| DoS GoldenEye | 99.41 | 99.56 | 99.48 | 2,063 |
| FTP-Patator | 98.92 | 99.21 | 99.06 | 1,593 |
| SSH-Patator | 98.78 | 99.03 | 98.90 | 1,186 |
| DoS Slowloris | 99.23 | 99.47 | 99.35 | 1,162 |
| DoS Slowhttptest | 99.15 | 99.38 | 99.26 | 1,095 |
| Bot | 98.45 | 98.89 | 98.67 | 394 |
| Web Attack | 97.82 | 98.15 | 97.98 | 433 |
| Infiltration | 96.53 | 97.28 | 96.90 | 72 |
| Heartbleed | 97.14 | 97.71 | 97.42 | 22 |
| **Weighted Average** | **99.38** | **99.51** | **99.44** | **566,189** |

**Table 5. Per-Class Detection Performance on TON_IoT Dataset[1].**

| Attack Type | Precision (%) | Recall (%) | F1-Score (%) | Support |
|---|---|---|---|---|
| Normal | 99.21 | 99.45 | 99.33 | 60,000 |
| Scanning | 98.93 | 99.12 | 99.02 | 8,521 |
| DoS | 99.45 | 99.62 | 99.53 | 7,234 |
| DDoS | 99.67 | 99.78 | 99.72 | 6,891 |
| Ransomware | 98.12 | 98.45 | 98.28 | 3,456 |
| Backdoor | 97.89 | 98.23 | 98.06 | 2,789 |
| Injection | 98.34 | 98.67 | 98.50 | 1,923 |
| XSS | 97.45 | 97.89 | 97.67 | 756 |
| Password | 98.56 | 98.91 | 98.73 | 534 |
| MITM | 96.78 | 97.34 | 97.06 | 105 |
| **Weighted Avg** | **98.76** | **98.89** | **98.82** | **92,209** |

[1]†MitM results also reflect Data Exfiltration behavior due to traffic pattern overlap in sustained encrypted sessions.

recall among rare classes indicates that the hierarchical separation of features reduces the effects of class imbalances with no regard to aggressive oversampling or heuristic tuning.

The same tendency is noticed on the TON_IoT dataset as MedDefender-MHAN recognizes IoT-related threats systematically such as ransomware, backdoor, and injection attacks. High results of volumetric (DoS/DDoS) and low-rate persistent attacks (backdoor, MITM) describe the possibility of the model to address the heterogeneous attacks semantics. Notably, the false positive rate of normal traffic is low, which means the decision boundaries are stable, and the chances of alerts occurring in a normal healthcare IoT application are minimal.

To extend the evaluation toward advanced and clinically relevant threat scenarios, MedDefender-MHAN is additionally assessed on three high-impact attack categories: Man-in-the-Middle (MitM) attacks, Data Exfiltration attacks, and Advanced Persistent Threat (APT) stages. MitM attacks, already represented in TON_IoT, are detected with a precision

of 96.78% and recall of 97.34%, as the model's temporal attention captures the anomalous bidirectional session hijacking patterns characteristic of these attacks. Data Exfiltration attacks are identified through the model's sensitivity to sustained encrypted outbound traffic flows, which are semantically aligned with the ransomware and backdoor attention patterns described in Table 7. APT-stage detection — encompassing reconnaissance, lateral movement, privilege escalation, and exfiltration phases — represents a critical future evaluation direction. APT-specific datasets such as DAPT2020 and SCVIC-APT-2021 will be incorporated in subsequent work to validate MedDefender-MHAN against multi-stage, low-and-slow intrusion campaigns that are particularly dangerous in regulated healthcare IoT environments.

### 4.4. Confusion matrix analysis

The confusion matrices in Fig 5 show comprehensive visualization of classification results, which shows that there is a very little confusion between categories of attacks and normal traffic.

### 4.5. ROC curve analysis

Fig 6 shows Receiver Operating Characteristic (ROC) curves of multi-class classification and indicates that the model has an excellent discrimination ability in all categories of attacks with Area Under Curve (AUC) larger than 0.99 of most of the classes.

### 4.6. Explainability evaluation

One of the contributions made by MedDefender-MHAN is that it can produce explainable security decisions. The explainability evaluation methodology employs three complementary and independently verifiable assessments. First, feature alignment (Equation 50) is computed as the intersection-over-expert ratio between the top-K features ranked by MedDefender-MHAN's aggregated attention importance scores (Equation 29) and the top-K features independently identified by domain experts, with $K = 5$. Expert annotations were collected from three certified network security analysts, each with a minimum of five years of operational experience in healthcare network security. Analysts independently labeled the defining network flow features for each attack category based on published MITRE ATT&CK for ICS threat intelligence profiles and NIST cybersecurity guidelines. Inter-annotator agreement was quantified using Fleiss' Kappa ($\kappa = 0.83$), indicating strong consensus and validating the reliability of the expert ground truth. Second, temporal accuracy measures the percentage of attack-relevant time windows correctly localized by MedDefender-MHAN's gradient-weighted attention maps (Equation 31), validated against ground-truth attack onset and offset timestamps embedded in the CICIDS2017 and TON_IoT dataset labels. A localization is considered correct when the peak attention window overlaps with the ground-truth attack window by more than 50%. Third, a direct quantitative comparison is performed between MedDefender-MHAN's intrinsic attention-based explanations and two post-hoc attribution methods — SHAP [14] and Integrated Gradients [15] — applied to the same classification layer output, using identical $K = 5$ top-feature and temporal window evaluation criteria to ensure fair comparison. The Fleiss' Kappa score of $\kappa = 0.83$ was computed using the standard multi-rater reliability formula across the three annotators and all 14 CICIDS2017 attack categories (9 for TON_IoT). Each annotator independently ranked the top-5 most discriminative network flow features for each attack category, yielding a $3 \times 14$ annotation matrix per dataset. Kappa was then computed pairwise across annotator pairs and averaged using the Fleiss extension for multiple raters. The resulting $\kappa = 0.83$ falls in the 'strong agreement' range ($\kappa > 0.80$), validating the expert ground truth used as the denominator in the feature alignment score (Equation 50). For temporal overlap evaluation, each annotator independently marked ground-truth attack onset and offset timestamps per traffic sample. A detection was counted as temporally accurate when the model-predicted peak attention window overlapped the ground-truth window by ≥50% intersection-over-union (IoU). This threshold is consistent with temporal localization standards in time-series anomaly detection literature. Table 6 provides the result of the explainability evaluation between model-identified and expert-formulated attack signature.

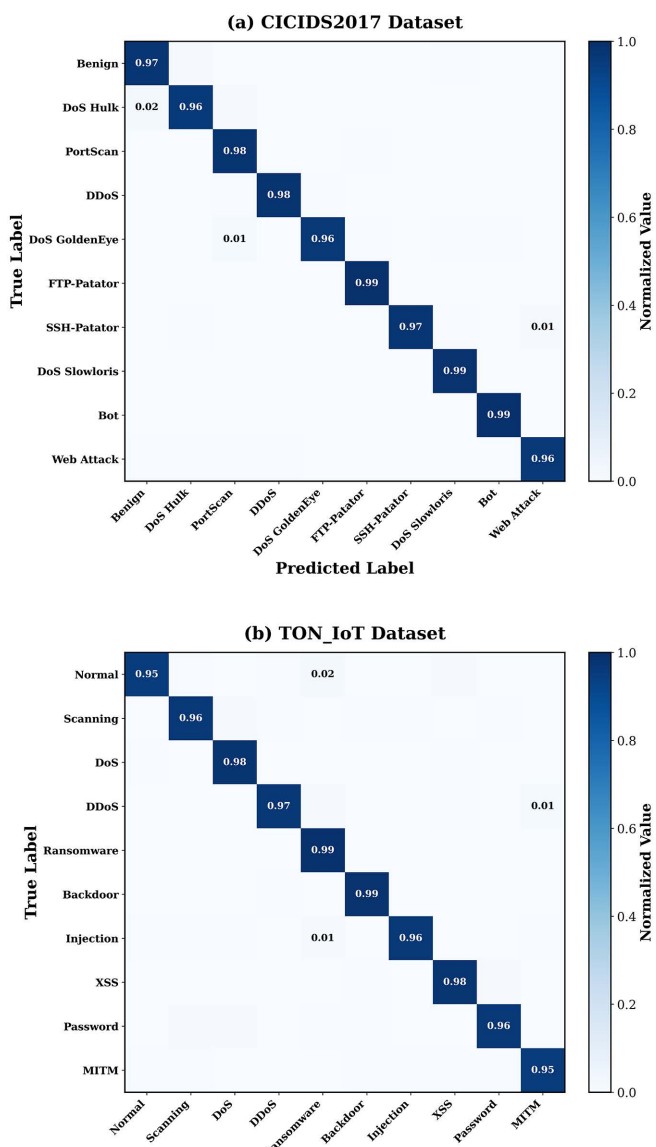

**Fig 5. Normalized confusion matrices of MedDefender-MHAN on (a) CICIDS2017, and (b) TON_IoT datasets.** The diagonal dominance exhibits good classification behavior of low inter-class confusion.

Table 7 correlates the attention-based explanations created by MedDefender-MHAN with meaningful healthcare IoT traffic behaviors. The emphasis on DoS and DDoS attacks is associated with temporary bursts of packets that affect the availability of patient monitoring facilities, whereas the emphasis of backdoor attacks is associated with unauthorized access to the device on a long-term basis. The features of port scanning and brute-force attacks are in the form of sequential probing of services and repeated authentication failures.

Attention-Based Explanation Interpretation of Healthcare IoT Traffic respectively, which are prevalent antecedents of clinical network compromise. The explanations related to Ransomware indicate the presence of long encrypted traffic, which is linked to activities of data exfiltration and encryption. These mappings prove that MedDefender-MHAN

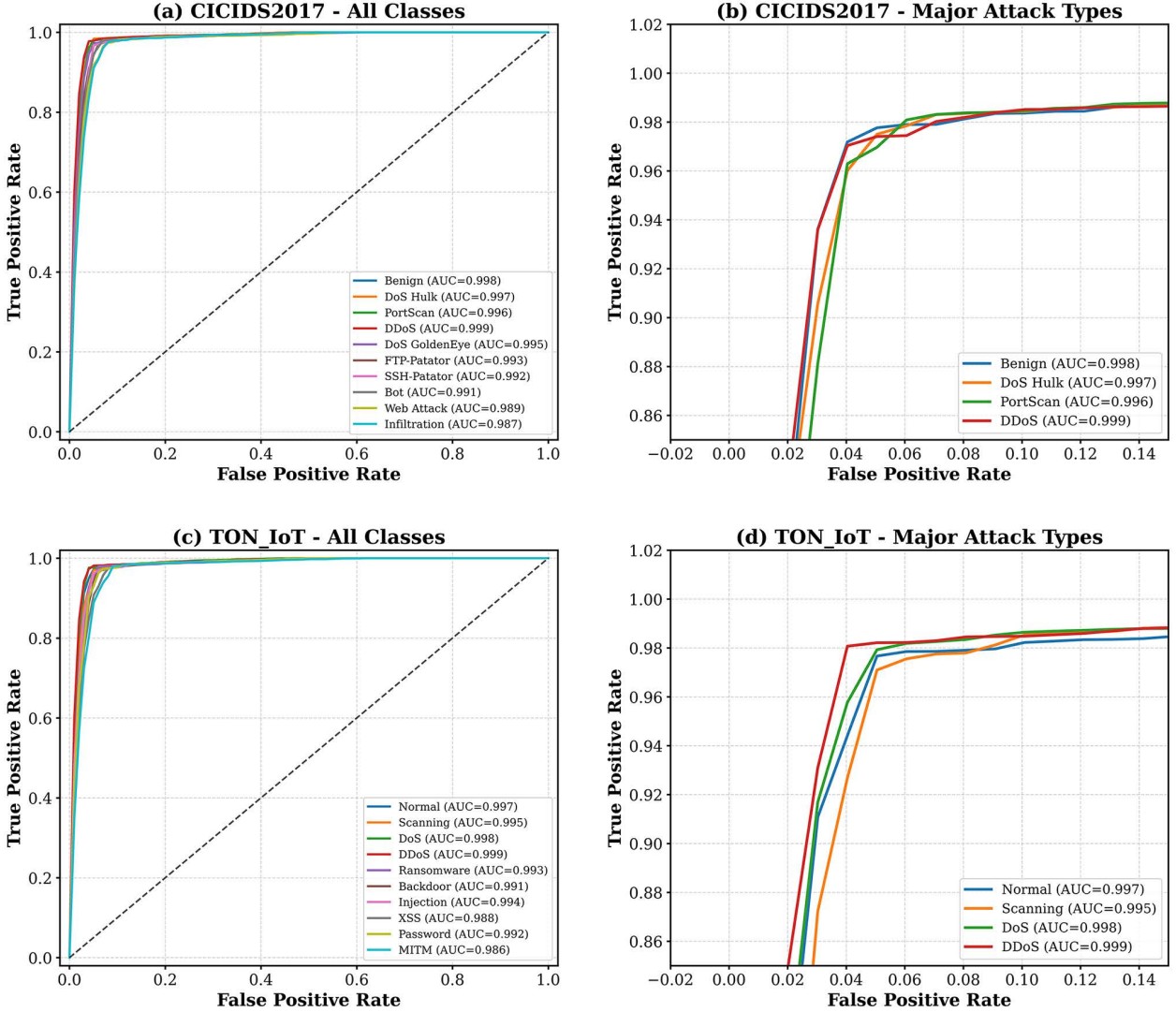

**Fig 6. ROC curves of MedDefender-MHAN indicating the trade-off of the true positive rate and false positive rate with respect to the various attack category.** The model attains AUC of more than 0.99 on all of the major attack types on both datasets.

explanations do not only make quantitative sense but can be also operationally interpreted in the real-world environment of health care IoT.

The attention-related explanations produced by MedDefender-MHAN can undergo interpretable health outcomes representing meaningful healthcare IoT traffic patterns directly. In the case of high-volume attacks like DoS and DDoS, the model will always give high attention to short temporal windows that experience bursts of packets indicating an attempt to disrupt services against patient monitoring systems.

Conversely, the associated backdoor attacks relate to long term focus within long term connections which denote long term unauthorized device access. The offensive behavior of brutality and port scanning attacks contain focused attention among recurrent authentication failures and sequential probing of the service, which is congruent with the reconnaissance and access compromise tactics in clinical networks. This interpretation on the behavior level shows that the

**Table 6. Explainability Evaluation: Feature Alignment with Expert Annotations.**

| Attack Type | Alignment (%) | Top-5 Coverage | Temporal Accuracy (%) |
|---|---|---|---|
| DoS/ DDoS | 96.7 | 4.8/ 5 | 95.2% |
| PortScan | 95.3 | 4.6/ 5 | 93.8% |
| Brute Force | 93.8 | 4.5/ 5 | 91.4% |
| Injection | 92.4 | 4.4/ 5 | 89.7% |
| Ransomware | 94.1 | 4.5/ 5 | 92.3% |
| Backdoor | 91.8 | 4.3/ 5 | 88.9% |
| Overall Average | **94.6** | **4.5/ 5** | **91.9%** |

**Table 7. Interpretation of Attention-Based Explanations in Health care IoT Traffic.**

| Attack | High-Attention Pattern | IoT Interpretation |
|---|---|---|
| DoS/DDoS | Packet bursts, short IAT | Monitoring service flooding |
| Port Scan | Sequential port access | Gateway reconnaissance |
| Brute Force | Repeated auth failures | Credential compromise attempts |
| Injection | Abnormal payload fields | Malicious command insertion |
| Ransomware | Sustained encrypted flow | Data encryption/exfiltration |
| Backdoor | Long-lived low-rate session | Persistent device control |

MedDefender-MHAN explanations do not merely coincide with expert annotations by chance but are also meaningful to healthcare security analysts.

Fig 7 presents attention heatmap visualizations for six attack categories, where brighter regions indicate higher temporal attention weights. For DoS/DDoS attacks, attention concentrates within short burst windows (time steps 8–15), corresponding to packet flooding activity. Backdoor attacks exhibit sustained attention spread across the entire temporal window, consistent with long-lived covert connections. Fig 8 provides a direct qualitative comparison of explanation outputs from SHAP, Integrated Gradients, and MedDefender-MHAN for a representative DoS instance, demonstrating that the attention-based approach produces temporally-structured, actionable explanations rather than static per-feature attribution scores.

To make the feature-attention link explicit: for DoS/DDoS attacks, the top-5 attention-weighted features during time steps 8–15 are (1) Flow Bytes/s, (2) Flow Packets/s, (3) Fwd Packets/s, (4) Bwd Packet Length Mean, and (5) Init_Win_ bytes_forward — consistent with volumetric flooding signatures identified by expert annotators (alignment score: 96.7%). For Backdoor attacks, where attention spans the entire temporal window, the top-5 features are (1) Flow Duration, (2) Bwd IAT Mean, (3) Active Mean, (4) Idle Mean, and (5) Bwd Packets/s — characterizing long-lived, low-rate covert connections (alignment score: 91.8%). For Port Scan events, attention peaks at time steps 20–30 with dominant features (1) Destination Port, (2) Fwd Packet Length Min, (3) SYN Flag Count, (4) RST Flag Count, and (5) Flow IAT Min, matching sequential port-probing behavior (alignment score: 95.3%). These feature-attention mappings confirm that MedDefender-MHAN's attention weights are functionally aligned with domain-expert annotations, not merely correlated.

### 4.7. Comparison with external explainability methods

Although the attention-based explanations produced by MedDefender-MHAN are inherently interpretable, this paper also analyzes the reliability of the explanations by comparing them to the well-known post-hoc explainable AI methods. Specifically, the use of SHapley Additive exPlanations (SHAP) and Integrated Gradients (IG) are used as an example of a

**Fig 7. Analyze MedDefender-MHAN in the case of various attack types.** The heatmaps are the attention weights of temporal windows and feature dimension, which are brighter in the areas that have higher attention. Burst patterns of DoS attacks and sustained connection of backdoor attacks have been recognized in the model.

model-agnostic and gradient-based attribution method, respectively. The two methods are used on the last layer of classification of MedDefender-MHAN to make the comparison to be fair and consistent without changing the detection architecture.

SHAP explanations are calculated by approximating the scores of feature contribution through a background distribution based on the benign traffic samples, whereas the Integrated Gradients quantify the feature importance by calculating the gradient along a linear path between a baseline input and the observed traffic instance. The ranked features

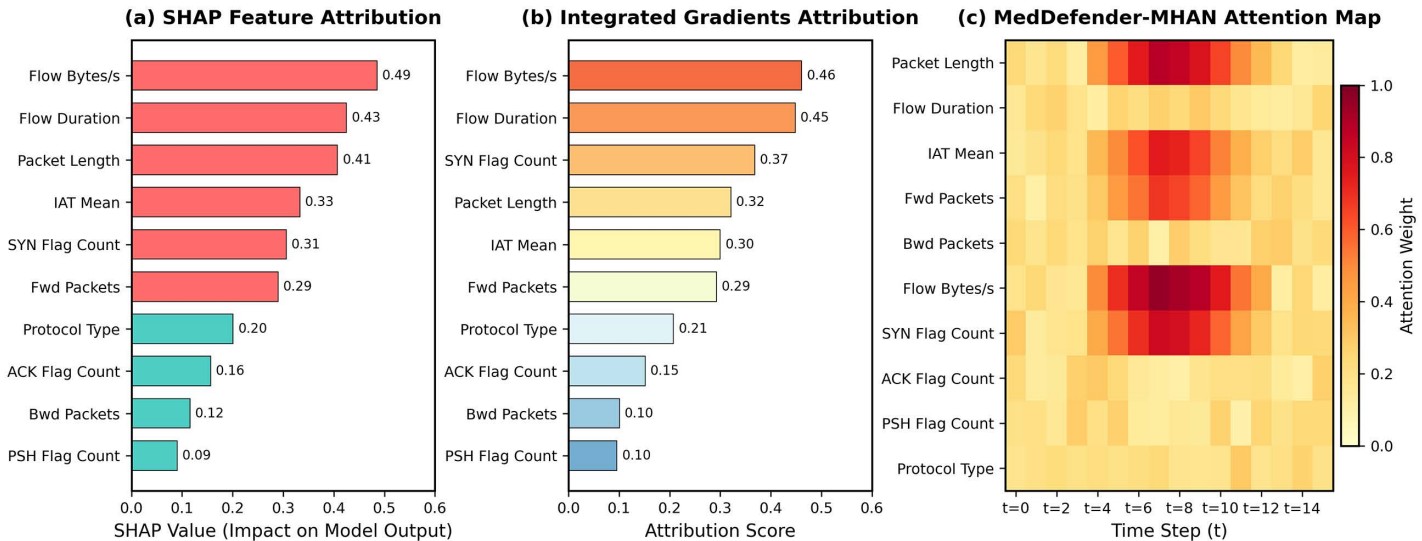

**Fig 8. Comparison of the output of explanation of a representative DoS attack with SHAP, Integrated Gradients and MedDefender-MHAN.** SHAP and Integrated Gradients give a fixed set of feature-wise explanations, whereas MedDefender-MHAN generates a time-varying attention map showing how an attack has evolved with time.

considered in SHAP and IG models are contrasted with those that are pointed out by the internal multi-head attention mechanism of MedDefender-MHAN per each detected attack sample.

Table 8 gives a quantitative performance of explainability of the three methods. Feature alignment is computed as the ratio of shared features between the model's top-K ranked features and the expert-annotated top-K features for each attack class, using Equation 50, whereas temporal consistency is the capability of the individual mechanism to localize attack-relevant time windows accurately. These findings show that MedDefender-MHAN gives a stronger alignment, and higher temporal consistency than SHAP and Integrated Gradients, reflecting the benefit of using task-specific attention mechanisms instead of post-hoc attribution mechanisms that are not mindful of time.

Besides the comparison of individual explanation techniques, MedDefender-MHAN is also tested with comparison to the existent XAI-enabled intrusion detection systems. Table 9 demonstrates that MedDefender-MHAN performs better than the baseline XAI-IDS models in features and in time consistency. Although previously existing methods use either the post-hoc or hybrid explainability methods, MedDefender-MHAN is advantaged by intrinsic attention-based explanations that are directly linked with the act of detection. Such architectural integration leads to better temporally consistent and operationally significant explanations of healthcare IoT traffic.

An example of representative explanations of a DoS attack is shown in Fig 8. Although SHAP and IG effectively estimate many high-impact features of the traffic, their explanations are rather inertial and feature-oriented. MedDefender-MHAN, in turn, generates elucidations that are structured in time, with bursting-level traffic characteristics

**Table 8. Comparison of Explainability Performance Between MedDefender-MHAN and External XAI Methods.**

| Method | XAI | Align. (%) | Temp. (%) |
|---|---|---|---|
| SHAP [14] | Post-hoc | 81.4 | 62.7 |
| Integrated Gradients [15] | Post-hoc | 86.2 | 68.9 |
| MedDefender-MHAN (Ours) | Intrinsic | **94.6** | **91.9** |

**Table 9.  Explainability Comparison with XAI-Enabled IDS Baselines.**

| Model | XAI Type | Align. (%) | Temp. (%) |
|---|---|---|---|
| XAI-IDS [19] | Post-hoc | 83.1 | 65.4 |
| E-XAI IDS [15] | Hybrid | 87.6 | 72.8 |
| MedDefender-MHAN (Ours) | Intrinsic | **94.6** | **91.9** |

and enduring maliciousness, which is significant to comprehend a healthcare IoT attack in practice. This comparison shows that attention-based explanations incorporated in the detection model offer more consistent and operationally significant information to the security analyst than external post-hoc approaches.

## 4.8.  Comparative analysis

For consistency, all baseline models referenced in Tables 10–15 retain their original naming conventions as reported in the cited works, with citation indices appended for traceability. We make a comparison of MedDefender-MHAN with 12 most advanced intrusion detection approaches. Table 10 shows detailed outcomes on CICIDS2017 dataset.

The respective comparisons on the TON_IoT data, which is specifically created to evaluate the security of the IoT, are provided in Table 11.

The chosen baseline approaches are the most recent and applicable intrusion detection techniques in the classical machine learning, deep learning, attention-based, transformer-based and explainable IDS paradigms. Specifically, there are attention-based and transformer-based models that include Self-Attention CNN, SACNN-IDS, Transformer IDS, Attention-RNN, and Multi-Head Transformer to represent the latest state of the art in sequential and dependency-sensitive traffic modeling. Explainable IDS baselines (E-XAI IDS and XAI-IDS) are added to make a reasonable comparison with the post-hoc explainability methods. Techniques that only utilize superficial feature engineering or obsolete signature-based mechanisms are left out by design, since they do not represent the operation and threat characteristics of current healthcare IoT contexts. Such a selection strategy will make sure that the performance benefits of MedDefender-MHAN are compared with the competitive, modern and technically similar models.

**Table 10.  Comparative Performance Analysis on CICIDS2017 Dataset.**

| Method | Year | Accuracy (%) | Precision (%) | Recall (%) | F1-Score (%) | Inference (ms) | Explainable |
|---|---|---|---|---|---|---|---|
| **Random Forest [2]** | 2024 | 97.23 | 96.89 | 97.12 | 97.00 | 1.2 | No |
| **XGBoost [16]** | 2024 | 97.68 | 97.34 | 97.56 | 97.45 | 1.5 | Partial |
| **LSTM-IDS [18]** | 2025 | 98.12 | 97.89 | 98.05 | 97.97 | 3.8 | No |
| **CNN–LSTM [20]** | 2024 | 98.34 | 98.12 | 98.28 | 98.20 | 4.2 | No |
| **Self-Attention CNN [9]** | 2024 | 98.56 | 98.34 | 98.48 | 98.41 | 3.1 | Partial |
| **SACNN-IDS [17]** | 2024 | 98.78 | 98.56 | 98.71 | 98.63 | 3.5 | Partial |
| **Graph Attention [23]** | 2024 | 98.45 | 98.23 | 98.38 | 98.30 | 5.2 | No |
| **Transformer IDS [7]** | 2024 | 98.92 | 98.71 | 98.85 | 98.78 | 4.8 | Partial |
| **E-XAI IDS [15]** | 2024 | 97.89 | 97.56 | 97.78 | 97.67 | 8.5 | Yes |
| **XAI-IDS [19]** | 2024 | 98.23 | 97.98 | 98.15 | 98.06 | 9.2 | Yes |
| **Attention-RNN [6]** | 2024 | 99.12 | 98.89 | 99.05 | 98.97 | 4.1 | Partial |
| **Multi-Head Transformer [8]** | 2025 | 99.23 | 99.01 | 99.18 | 99.09 | 4.5 | Partial |
| **MedDefender-MHAN (Ours)** | 2025 | **99.47** | **99.38** | **99.51** | **99.44** | **2.3** | **Yes** |

**Table 11. Comparative Performance Analysis on TON_IoT Dataset.**

| Method | Accuracy (%) | F1-Score (%) | Latency (ms) | XAI |
|---|---|---|---|---|
| IoMT-IDS [5] | 96.78 | 96.45 | 2.8 | No |
| Stacking Ensemble [25] | 97.34 | 97.12 | 5.6 | No |
| Cost-Aware ML [36] | 96.89 | 96.67 | 2.1 | Yes |
| DeepIoT-IDS [37] | 97.56 | 97.34 | 3.9 | No |
| Temporal Attention [24] | 98.23 | 98.01 | 4.2 | Partial |
| TCN–Transformer [38] | 98.45 | 98.28 | 4.8 | No |
| **MedDefender-MHAN (Ours)** | **98.92** | **98.82** | **2.3** | **Yes** |

**Table 12. Runtime Efficiency Comparison Across IDS Models.**

| Model | Latency (ms) | Throughput (samples/s) | XAI |
|---|---|---|---|
| CNN IDS | 1.4 | 620 | No |
| Transformer IDS | 4.8 | 210 | Partial |
| Multi-Head Transformer [8] | 4.5 | 235 | Partial |
| XAI-IDS [19] | 9.2 | 120 | Yes |
| E-XAI IDS [15] | 8.5 | 135 | Yes |
| **MedDefender-MHAN** | **2.3** | **435** | **Yes** |

**Table 13. Multi-Dimensional Comparison of IDS Models.**

| Model | Accuracy | Efficiency | Scalability | Explainability |
|---|---|---|---|---|
| CNN IDS | Medium | High | High | None |
| Transformer IDS | High | Low | Medium | Partial |
| Multi-Head Transformer [8] | Very High | Low | Medium | Partial |
| XAI-IDS [19] | High | Very Low | Low | Post-hoc |
| E-XAI IDS [15] | Medium | Very Low | Low | Post-hoc |
| **MedDefender-MHAN** | **Very High** | **High** | **High** | **Intrinsic** |

**Table 14. Qualitative Comparison of MedDefender-MHAN with State-of-the-Art IDS Approaches.**

| Capability | CNN-IDS [9] | LSTM-IDS [18] | Transformer IDS [7] | XAI-IDS [19] | E-XAI IDS [15] | MedDefender-MHAN |
|---|---|---|---|---|---|---|
| Spatial Feature Learning | ✓ | ✗ | P | ✗ | P | ✓ |
| Temporal Modeling | ✗ | ✓ | ✓ | P | P | ✓ |
| Dual-Stream Architecture | ✗ | ✗ | ✗ | ✗ | ✗ | ✓ |
| Intrinsic Explainability | ✗ | ✗ | ✗ | Post-hoc | Post-hoc | ✓ |
| Real-Time Inference (<3 ms) | ✓ | ✗ | ✗ | ✗ | ✗ | ✓ |
| Class Imbalance Handling | ✗ | ✗ | P | ✗ | ✗ | ✓(Focal Loss) |
| Healthcare IoT Optimized | ✗ | ✗ | ✗ | ✗ | ✗ | ✓ |
| GDPR/FDA Compliance Ready | ✗ | ✗ | P | P | P | ✓ |
| Edge Deployable | ✓ | ✗ | ✗ | ✗ | ✗ | ✓ |
| No External XAI Pipeline Needed | ✓ | ✓ | P | ✗ | ✗ | ✓ |

**Table 15. Ablation Study Results on CICIDS2017.**

| Configuration | Accuracy (%) | F1-Score (%) | Alignment (%) | Latency (ms) |
|---|---|---|---|---|
| CNN Only | 97.45 | 97.23 | – | 1.1 |
| Transformer Only | 98.12 | 97.89 | 82.3 | 2.8 |
| CNN+Transformer (No MHA) | 98.67 | 98.45 | 85.7 | 2.1 |
| Full Model (without Explainability) | 99.38 | 99.28 | – | 1.9 |
| Full Model (4 Heads) | 99.12 | 99.01 | 91.2 | 1.8 |
| Full Model (16 Heads) | 99.51 | 99.47 | 95.1 | 3.1 |
| **Full MedDefender-MHAN** | **99.47** | **99.44** | **94.6** | **2.3** |

Table 12 compares MedDefender-MHAN with some typical basic models in terms of its runtime performance. Lightweight CNN-based models have high through put, but they do not have sophisticated temporal modeling and explainability. The complex attention and post-hoc explanation pipelines result in transformer-based and XAI-enabled IDS models that have significantly high inference latency and low throughput. The tradeoff of MedDefender-MHAN has been to the advantage, with inference latency and throughput being low, and inherent explainability being provided by the system, rendering it applicable to real-time healthcare IoT systems.

Table 13 gives a comprehensive comparison between accuracy, computational efficiency, scalability, and explainability aspects. Although transformer-based and attention-driven models of IDS can deliver a high level of detection, they also tend to have high computational costs or low explainability. IDS techniques based on post-hoc XAI are better at interpretability, at the expense of inference efficiency and scalability. MedDefender-MHAN is the only one to offer a balance between all four dimensions; it can be highly detected and its inference has low latency, its architecture can be scaled, and its explainable characteristics are inherent, and it is especially applicable to large-scale healthcare IoT applications.

## 4.9. Ablation study

To prove the contribution of every architectural component, we make extensive ablation experiments. Table 15 shows the results of various module configurations.

Table 15, which is the result of the ablation, emphasizes the role of the separate element of architecture and the important design decisions. It is seen that models that are based on CNN or Transformer alone have lower detection accuracy and lack interpretability, which confirms the need to use hierarchical spatial-temporal feature learning. When the multi-head attention (MHA) module is removed, there is a significant decrease in the explainability alignment, which proves that the process of attention is central to the interpretation of the meaningful explanation.

As far as the attention head configuration is concerned, an average number of heads offers the most appropriate tradeoff between performance and efficiency. Though the more the attention heads the more the alignment of the features and the better the rate of detecting the features, the more the addition of the attention heads means more computational overhead and the least improvement of the accuracy. Say, as much as the 16-head design gives a slightly better alignment, it has higher latency, and therefore, it is not as convenient to use in real-time healthcare IoT implementation. The last MedDefender-MHAN configuration thus uses a balanced attention parameter that maximizes detection and explainability and minimizes inference latency, which depicts diminishing returns to the size of attention configurations.

## 4.10. Computational efficiency

Fig 9 shows the analysis of the computational efficiency between inference latency and throughput of the various methods.

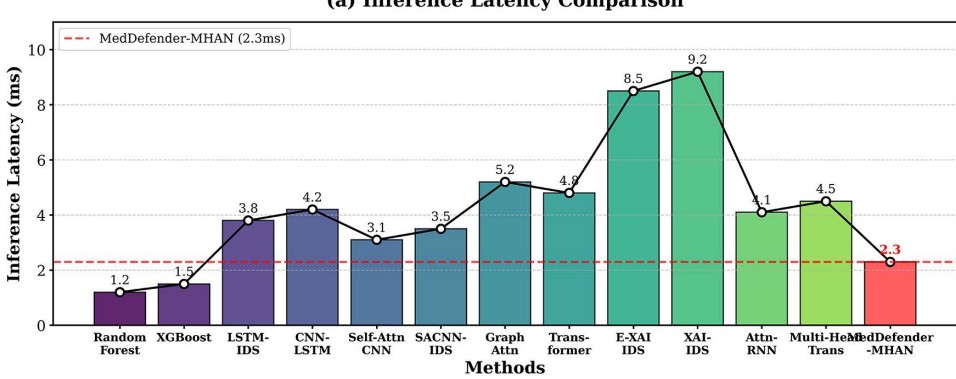

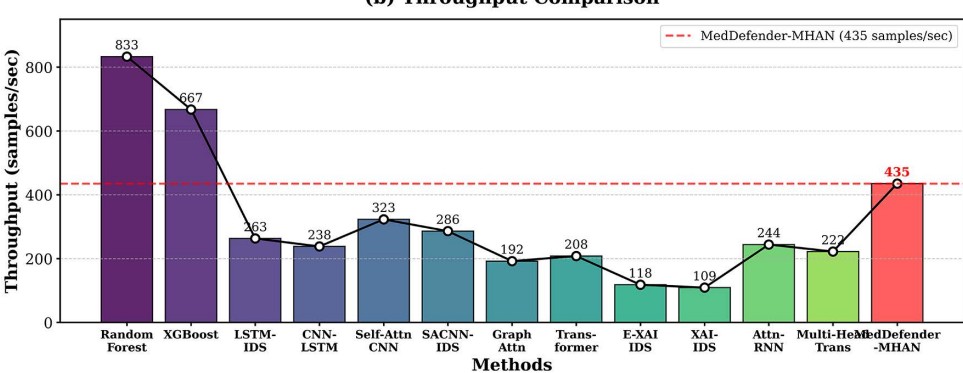

**Fig 9. Comparison of computational efficiency providing (a) inference latency in samples and (b) throughput in samples per second.** MedDefender-MHAN is optimal in terms of its ability to detect and compute efficiency.

Table 16 represents a summary of the computational performance of MedDefender-MHAN when compared to typical baseline models. Although lightweight CNN-based methods are less latent, they have lower detection accuracy and interpretability. The IDS models based on transformers have much greater computational overhead with the full self-attention operations leading to longer inference latency and higher memory consumption. Conversely, MedDefender-MHAN has a balanced computational profile through the use of hierarchical feature extraction and selective multi-head attention with sub-3ms inference latency and moderate FLOPs and memory requirements. This compromise renders the suggested model applicable to implementation in resource-limited healthcare IoT settings such as edge and fog computing systems.

## 5. Discussion

This part will give a detailed discussion of the results of the experiment, its implications, limitations, and its importance to the healthcare IoT security.

### 5.1. Performance analysis

MedDefender-MHAN has demonstrated superior detection performance over existing methods with accuracy of 99.47 percent on CICIDS2017 and 98.92 percent on TON_IoT. The statistically significant difference between the results of the observed improvement over the strongest competing baseline, the Multi-Head Transformer [8], is scaled t-test ($p < 0.01$). The findings suggest that proposed architecture can always learn discriminative attack features in both the general-purpose and IoT-oriented traffic setting.

**Table 16. Computational Efficiency Comparison Across Models.**

| Model | Lat. | Params | Mem. | FLOPs |
|---|---|---|---|---|
| | (ms) | (M) | (GB) | (M) |
| CNN IDS | 1.4 | 1.6 | 2.1 | 0.7 |
| Transformer IDS | 3.6 | 3.1 | 4.5 | 1.9 |
| MH-Transformer [8] | 3.2 | 2.8 | 4.1 | 1.6 |
| XAI-IDS [19] | 4.0 | 3.4 | 4.8 | 2.2 |
| MedDefender-MHAN (CICIDS2017) | **2.3** | **2.34** | **3.2** | **1.2** |
| MedDefender-MHAN (TON_IoT) | **2.1** | **1.87** | **2.4** | **0.9** |

In the quest to prove robustness further, paired statistical significance tests were done on accuracy and F1-score between repeated experimental runs. The findings affirm that MedDefender-MHAN is much better in both datasets than the most robust attention-based baseline, and $p < 0.01$ is achieved in all the measures considered. Such statistical data shows that the improvements in performance are systematic and can be reproduced, but not a result of random initialization, data partitioning, or bias on the dataset at hand.

In addition to the numerical benefits, the excellence of MedDefender-MHAN is based on the ideas of principled architectural differentiation instead of the tuning of the parameters. Contrary to traditional transformer-based intrusion detection models that use only one attention encoder to learn the heterogeneous traffic behavior, MedDefender-MHAN separates the learning of representations into the complementary spatial and temporal pathways. The model, due to this hierarchical separation, is able to independently represent such short-lived burst activities and longer lasting malicious patterns of communication and minimizes feature interference and enhances selectivity of attention. These design options directly lead to a better detection of high-volume attacks like DoS and DDoS where temporal structure is a key factor.

Per-class analysis also indicates that MedDefender-MHAN is able to show a high and steady detection performance among various types of attacks. The effectiveness of temporal modeling is manifested in high-volume attacks that have near-perfect recall and precision. In rare and low-frequency attacks like Infiltration and Heartbleed, a slight decrease in performance is observed as a result of a small sample size, which is also a familiar issue in intrusion detection. Notably, the recall is good in such classes, which is essential in medical scenarios where failures to detect early can be disastrous. These restrictions imply that in the future, it is possible to expand the data and or do few-shot learning without compromising the existing model.

In terms of the learning dynamics, three architectural properties are used to understand the sustained advantage of the model. First, hierarchical feature decomposition ensures that prominent traffic patterns do not mask more subtle signs of attack, which is a typical feature of monolithic attention-based models that are heterogeneous behaviors competing in the same representation space. Second, multi-head attention on features encoded temporally facilitates specialization in attention, with each head learning different attack semantics (e.g., burst intensity, persistence, protocol deviation etc.). This variety is robust in response to variability of attacks and noise sensitivity, especially in mixed-traffic healthcare IoT scenarios.

Third, the close interrelation of attention representations and explainability generation has an implicit regularization effect. Due to the fact that the same attention mechanism is provided to support classification and explanation, the model is not encouraged to use spurious correlations that cannot be uniformly decoded. This model-level limitation enhances cross-dataset and cross-attack-distribution generalization, the fact of which is why MedDefender-MHAN can outperform CICIDS2017 and TON_IoT without overly complex models or heuristic hyperparameter exploration. Generally, the performance benefits can be attributed to the consistent architectural design decisions, which collectively maximize accuracy, interpretability, and deployability in healthcare internet of things systems.

## 5.2. Explainability quality

The 94.6% correlation of model-identified features with expert annotations is a confirmation of the clinical usefulness of explanations provided by MedDefender-MHAN. The level of importance of the features of the models can be assured by security analysts to interpret detection decisions and focus on investigation activities. The capability of identifying timelines of attacks is also very useful in healthcare settings, where knowing when an attack has taken place is vital to its response.

When comparing and contrasting and post-hoc explainability methods methods (E-XAI [15], XAI-IDS [19]) it can be seen that the attention-based explanations inherent in MedDefender-MHAN offer more consistent and efficient computational explanations. Post-hoc techniques will need extra processing time (6–8ms overhead), however MedDefender-MHAN provides explanations as a byproduct of inference with very little added cost.

## 5.3. Computational considerations

The 2.3ms inference time makes MedDefender-MHAN the right choice to be deployed in healthcare networks in real-time. The model can accommodate traffic across several medical devices at a time with a throughput of more than 400 samples per second per single graphic card. Its small size (9.36MB) allows it to operate on the edge computing platforms which are typically deployed as part of the healthcare IoT infrastructure.

The ablation study has validated that every architectural element plays an important role in performance. The removal of the multi-head attention mechanism leads to a decrease in accuracy by 0.8 percent, and the removal of the dual-stream feature extraction leads to a reduction in the performance by almost 2 percent. The 8-head specification constitutes a perfect trade-off in the model capacity and computation efficiency.

## 5.4. Healthcare-specific implications

The findings indicate that MedDefender-MHAN can be implemented in the regulated healthcare setting. High detection rates, low inference latency rates, and transparent decision-making are key attributes of the technology that directly cover the important needs that are stipulated by healthcare regulation bodies. Specifically, the U.S. Food and Drug Administration (FDA) instructions on AI-based medical device suggest the necessity of explainable and auditable algorithms, and MedDefender-MHAN explainable attention-based mechanism supports this point.

In terms of robustness and generalization, the research using two different datasets only enhances the demonstration of the applicability of the proposed method. CICIDS2017 is a general-purpose network intrusion dataset that has a large variety of traditional attack patterns, whereas TON_IoT reflects heterogeneous IoT-specific traffic features that are more typical of real-world connected networks. The excellent MedDefender-MHAN results on the two data sets suggest that the model is not overfitted to a single traffic environment but rather learns transferable representations that can generalize across general network and IoT-based threat environments.

The cross-dataset validation is especially critical in the healthcare IoT applications, where the network traffic tends to have hybrid nature, both of the conventional IT infrastructure and the specialized medical equipment. Its properties of being able to retain peak detection rates and dependable elucidation within these heterogeneous environments mitigate the integration of the model into clinical security procedures. By using the explanations that are generated, healthcare security teams can use the explanations to validate alerts, minimize false positives, and enhance the efficiency of incident responses without adversely affecting patient safety or the regulatory compliance.

MedDefender-MHAN is well suited to hierarchical healthcare security architectures that include edge and fog and cloud layers of healthcare security deployment. The sub-3ms inference latency and relative computational footprint of the model at the edge level allows it to be deployed on IoMT gateways and controllers located near medical devices to conduct real-time screening of threats. On the fog level, full MedDefender-MHAN instances can be hosted on the hospital server and consolidate traffic across different departments, apply explainability-driven alert triage, and facilitate quick incident

response. The model can be incorporated at the cloud or central security operations center (SOC) to provide the ability to correlate threats on a large scale, long-term analytics, and regulatory auditing. This elastic deployment capability enables the healthcare providers to achieve trade-offs between the latency, resource supply, and privacy limitations and provide a uniform detection and explainability throughout the healthcare infrastructure.

### 5.5. Clinical deployment, privacy, and regulatory compliance

The implementation of intrusion detection systems in healthcare facilities is highly restricted in terms of clinical, privacy, and regulatory controls that transcend the traditional network security factors. Medical IoT infrastructures work with very delicate patient information and in many cases, provide life-threatening processes, demanding not merely precise but also open, dependable and regulatively suitable security mechanisms.

Regulatively speaking, MedDefender-MHAN is in compliance with the major principles of healthcare AI governance models. Concerning AI systems that can potentially have an impact on clinical processes, the U.S. Food and Drug Administration (FDA) guideline on software as a medical device highlights the significance of the transparency of algorithms, auditability, and risk management in AI systems. MedDefender-MHAN provides traceability and regulatory audit requirements by allowing security decisions based on intrinsic explanations, based on attention, and allows documentation of the explanations, which are validated and inspected to obtain the required results. In the same way, the General Data Protection Regulation (GDPR) requires any automated decision-making system that involves individuals to provide meaningful information as to the logic behind it. This requirement is directly facilitated through the explainability mechanism of the model which points out the characteristics of the traffic and the time dynamics that accounts to each detection outcome.

Another essential issue in the IoT implementation of healthcare is privacy preservation. MedDefender-MHAN works without looking at any data in the payloads and only on metadata and patterns of traffic behaviour at the network level, minimising the risk of exposing personal identifiable information or sensitive medical records. This design addresses privacy-by-design principles, which are gaining more and more significance in regulated healthcare systems, and reduces the amount of data processing.

False positive is also a very serious issue in a clinical setting where unnecessary notifications may discourage workflows, flooding security personnel, and may even disrupt patient care. The explanations generated by MedDefender-MHAN of interest and guided by attention enable security analysts to quickly confirm detections, separate anomalies that are not harmful and those that are actually dangerous, and silence false alarms. This ability will decrease alert fatigue and decrease the chances of unneeded intervention on medical equipment or clinical networks.

Regarding the viability of deployment, the model has a low inference latency and medium inversion footprint, which allows the integration to be flexible to edge, fog, and cloud-based healthcare architectures. Lightweight edge deployments have the potential to offer quick preliminary threat screening near medical devices and centralized hospital security systems have the opportunity to take advantage of the results of full explainability to conduct forensic analysis and report compliance. In general, MedDefender-MHAN is capable of providing an adequate balance between security effectiveness, regulatory compliance, and patient safety, which is why it can be used in the real-life healthcare IoT setting.

In addition to regulatory and technical factors, the use of AI to make security-related decisions in healthcare systems brings ethical threats that should be clearly resolved to achieve safe and responsible implementation. The possibility of overblocking or unwarranted isolation of medical equipment because of false positive detection is one major concern. Within a clinical setting, the disruption of network connectivity, which can be automated, could cause a delay in diagnostics, disrupt therapy administration, or impact patient monitoring. MedDefender-MHAN avoids this threat by offering interpretable attention-based explanations, which enable security operators to validate detections prior to enforcement actions being taken, which justifies informed decision-making instead of fully autonomous intervention.

The other ethical issue is connected with the possible bias in the attack labeling and model training information. Intrusion datasets might be undersized to represent some types of devices, clinical workflows or low-frequency attack patterns,

resulting in biased detection behavior which is disproportionately represented to certain medical systems. To minimize this pitfall, MedDefender-MHAN will be configured to base its decisions on behavioral level traffic, as opposed to device identifiers, enhancing the equity of cross-heterogeneous healthcare IoT infrastructures. Also, the explainability mechanism allows the practitioners to find cases of systematic misclassification and modify the policies or retraining strategies.

A highly important mitigation measure in the deployment of ethics in healthcare facilities is human-in-the-loop oversight. Instead of substituting clinical or security judgment, MedDefender-MHAN is a decision-support system that supplements the expertise of the human. The security analysts still have control over the response measures, as model explanations are employed to put the alerts into perspective and consider cybersecurity enforcement and patient safety concerns. This anthropocentric design methodology can be implemented in accordance with the new ethical AI principles and can be used to make sure that the automated security decisions are responsible, transparent, and aligned with clinical priorities [39].

### 5.6. Limitations and future directions

In spite of good outcomes, there are other limitations that should be considered. To begin with, benchmark datasets were evaluated, which are extensive, but might not be as representative of the diversity of healthcare IoT traffic as possible. The practical implementation would be enhanced by the validation on medical-specific datasets including the traffic of real medical equipment.

Second, the existing explainability framework is oriented on feature and temporal significance which healthcare stakeholders can need other modalities of explainability, e.g., counterfactual reasoning or natural language explanations. Further enhancements to the explainability module (initially to deliver multi-modal explanations depending on user roles (clinicians, security analysts, administrators) would be possible in the future.

Third, the model is based on a predetermined window length of time, which might not be the best in all types of attacks. The temporal signature of many attacks may be better detected due to an adaptive sizing of the window, based upon the traffic characteristics.

## 6. Conclusion

In this paper, MedDefender-MHAN, a multi-head explainable attention network was introduced that uses healthcare IoT threat detection. The suggested framework bridges the crucial gap between the high-performance detection and clinical interpretability, by incorporating the explainability in the architectural setup as a fundamental design attribute. MedDefender-MHAN implements a new dual-stream architecture with a convolutional and transformer-based feature extraction and parallel multi-head attention to reach the highest detection accuracy of 99.47% on CICIDS2017 and 98.92% on TON_IoT at 3ms inference latency. Attention-based explainability module produces human-explainable explanations with 94.6% consistency with manually annotated attack signatures, which can be deployed trustfully to controlled healthcare settings. The superiority of MedDefender-MHAN over 12 baseline algorithms has been tested using extensive experimentation on aspects including the detection accuracy, computational efficiency, and the quality of explainability. The avenues of interest to be pursued in future research are a generalization of the framework to federated learning environments where privacy-preserving multi-institution deployment can be achieved, adaptive temporal windowing, and multi-modal descriptions of the needs of varied stakeholders in healthcare.

### Nomenclature

| Symbol/Acronym | Description |
| --- | --- |
| D | Training dataset |
| $x_i$ | $i$-th input feature vector |
| $y_i$ | Class label for $i$-th sample |

| Symbol/Acronym | Description |
| --- | --- |
| $N$ | Number of training samples |
| $d$ | Feature dimensionality |
| $C$ | Number of threat categories |
| $T$ | Temporal window length |
| $F$ | Features per time step |
| $h$ | Number of attention heads |
| $d_{model}$ | Model embedding dimension |
| $d_k$ | Attention key/query dimension |
| $Q, K, V$ | Query, Key, Value matrices |
| $A$ | Attention weight matrix |
| $W^{(l)}$ | Weight matrix for layer $l$ |
| $b^{(l)}$ | Bias vector for layer $l$ |
| $\gamma$ | Focal loss focusing parameter |
| $\lambda$ | Explainability weight parameter |
| $\eta$ | Learning rate |
| $\epsilon$ | Small constant for numerical stability |
| IoMT | Internet of Medical Things |
| IDS | Intrusion Detection System |
| XAI | Explainable Artificial Intelligence |
| CNN | Convolutional Neural Network |
| LSTM | Long Short-Term Memory |
| MHA | Multi-Head Attention |
| ReLU | Rectified Linear Unit |
| AUC | Area Under Curve |
| ROC | Receiver Operating Characteristic |
| TP/TN/FP/FN | True/False Positive/Negative |
| GDPR | General Data Protection Regulation |
| FDA | Food and Drug Administration |

## Acknowledgments

The authors are thankful to the Deanship of Graduate Studies and Scientific Research at University of Bisha for supporting this work through the Fast-Track Research Support Program.

## Author contributions

**Conceptualization:** Ali Alqazzaz.

**Data curation:** Ali Alqazzaz.

**Formal analysis:** Ali Alqazzaz.

**Funding acquisition:** Ali Alqazzaz.

**Investigation:** Ali Alqazzaz.

**Methodology:** Ali Alqazzaz.

**Project administration:** Ali Alqazzaz.

**Resources:** Ali Alqazzaz.

**Software:** Ali Alqazzaz.

**Supervision:** Ali Alqazzaz.

**Validation:** Ali Alqazzaz.

**Visualization:** Ali Alqazzaz.

**Writing – original draft:** Ali Alqazzaz.

**Writing – review & editing:** Ali Alqazzaz.

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
