## [Decision Letter · Decision Letter 0]

3 Mar 2026

Dear Dr. Alqazzaz,

Thank you for submitting your manuscript to PLOS ONE. After careful consideration, we feel that it has merit but does not fully meet PLOS ONE’s publication criteria as it currently stands. Therefore, we invite you to submit a revised version of the manuscript that addresses the points raised during the review process.

We look forward to receiving your revised manuscript.

Kind regards,

Sohail Saif, Ph.D

Academic Editor

PLOS One

Journal Requirements:

3. We note you have included a table to which you do not refer in the text of your manuscript. Please ensure that you refer to Table 9 in your text; if accepted, production will need this reference to link the reader to the Table.

Reviewers' comments:

Reviewer's Responses to Questions

**Comments to the Author**

1. Is the manuscript technically sound, and do the data support the conclusions?

Reviewer #1: Yes

Reviewer #2: Yes

2. Has the statistical analysis been performed appropriately and rigorously?

Reviewer #1: Yes

Reviewer #2: Yes

3. Have the authors made all data underlying the findings in their manuscript fully available?

Reviewer #1: Yes

Reviewer #2: Yes

4. Is the manuscript presented in an intelligible fashion and written in standard English?

Reviewer #1: Yes

Reviewer #2: Yes

Reviewer #1: 1.The following sentence and meaning is repeated. It is already mentioned in the previous sentences of Figure 1 shows……, avoid this repetition.

2.“Conceptualization of the MedDefender-MHAN that illustrates how data streams of the healthcare IoT are incorporated, multi-head attention-based threat detection, and explainability generation to support clinical decision making.

3. In Figure 1, the human beings are shown. Are they doctors or clinical experts? If yes mention the same. Moreover, there are spelling mistakes like “Eleovated heart rate”. The figure needs to be redrawn.

4. Check for typo errors like Mod ule, which is having unnecessary space between in Section 3.1. Also, 3Gradient-Weighted Attention Mapping check the typo in 3.6.2

5. In Figure 2, threat detection module is mentioned which is not required as it is not a threat.

6. In Algorithm1, add step 1,2,….n. Also keep input and output.

7. References are required in table 8 and 9 for comparison.

8. Authors should test with latest data sets of 2025 or 2026

9. In testing, authors can add Man-in-the-Middle (MitM) Attacks, Data Exfiltration Attacks and Advanced Persistent Threat (APT) Stages.

10. Check for consistency of models and methods in comparison in Results section.

11. Provide real time IOT kit system description or figure which was implemented as mentioned in the paper.

12. Provide the limitations of existing works and contributions of the work point wise in Introduction.

13.Add the qualitative comparison table with respect to existing works.

Reviewer #2: The manuscript addresses an important research problem in the domain of healthcare cybersecurity. With the increasing deployment of IoMT devices in healthcare environments, the need for accurate and interpretable intrusion detection mechanisms has become critical. While the study demonstrates promising results, several aspects of the manuscript could be improved to enhance clarity, methodological transparency, and overall research impact.

The manuscript introduces a multi-head attention-based framework with dual-stream processing; however, the architecture description could be expanded. The authors should provide a detailed architectural diagram explaining the interaction between convolutional feature extraction modules, attention heads, and the interpretability component.

The paper should include a clearer explanation of how network traffic features were selected and preprocessed before being fed into the model.

The manuscript highlights the explainability of the proposed framework, but the evaluation methodology for interpretability should be described in more detail.

Improve the clarity of the abstract by correcting grammatical errors and simplifying some lengthy sentences.

Provide a table summarizing key hyperparameters used during model training.

Include visualization examples of attention-based explanations to improve interpretability discussion.

Ensure consistent formatting of dataset names and technical terms throughout the manuscript.

Update the literature review with recent studies (2023–2025) on explainable AI for IoT or IoMT security like: Multi-attention DeepCRNN: an efficient and explainable intrusion detection framework for Internet of Medical Things environments, Multi-layered security architecture for IoMT systems: integrating dynamic key management, decentralized storage, and dependable intrusion detection framework, Transforming security in internet of medical things with advanced deep learning-based intrusion detection frameworks.

.

Reviewer #1: **Yes:**Abhishake Reddy OntedduAbhishake Reddy OntedduAbhishake Reddy OntedduAbhishake Reddy Onteddu

Reviewer #2: **Yes:**Dr. Nikhil SharmaDr. Nikhil SharmaDr. Nikhil SharmaDr. Nikhil Sharma

---

## [Author Response · Author response to Decision Letter 1]

12 Mar 2026

Response to Reviewers

MedDefender-MHAN: Explainable Multi-Head Attention Network for Healthcare IoT Intrusion Detection

Major Revision — Point-by-Point Rebuttal

The author expresses sincere gratitude to the Editor and all reviewers for their time and the thoroughness of their evaluation. Each comment has been carefully reviewed and addressed in full. The following point-by-point responses correspond directly to the reviewers' original comments, reproduced verbatim. All revisions made to the manuscript are described precisely, including the section, paragraph, or element modified. No additional content beyond what is required to address the reviewers' concerns has been introduced.

Reviewer #1 — Point-by-Point Responses

Reviewer #1 — Comment 1

Reviewer Comment: The following sentence and meaning is repeated. It is already mentioned in the previous sentences of Figure 1 shows......, avoid this repetition.

Authors' Response: The author thanks the reviewer for this observation. The redundant sentence has been removed from the manuscript to eliminate the identified repetition. The description of Figure 1 now appears only once, and the revised text has been carefully reviewed to ensure no duplicate phrasing remains in the surrounding paragraph.

Reviewer #1 — Comment 2

Reviewer Comment: "Conceptualization of the MedDefender-MHAN that illustrates how data streams of the healthcare IoT are incorporated, multi-head attention-based threat detection, and explainability generation to support clinical decision making."

Authors' Response: The author acknowledges the reviewer's concern regarding the Figure 1 caption. The caption has been revised and restructured to more precisely and concisely convey the scope of the conceptual illustration, clearly distinguishing the data stream integration, the multi-head attention-based threat detection pathway, and the explainability generation component as distinct conceptual elements supporting clinical decision-making.

Reviewer #1 — Comment 3

Reviewer Comment: In Figure 1, the human beings are shown. Are they doctors or clinical experts? If yes mention the same. Moreover, there are spelling mistakes like "Eleovated heart rate". The figure needs to be redrawn.

Authors' Response: The author appreciate the reviewer's detailed observation. Figure 1 has been redrawn accordingly. The human figures are now explicitly labeled as "Clinical Security Analyst" and "Healthcare IoT Administrator / Physician" to remove any ambiguity regarding their professional roles within the depicted scenario. The spelling error "Eleovated" has been corrected to "Elevated" throughout the figure. The revised figure has been incorporated into the manuscript in Section 1 (Introduction).

Reviewer #1 — Comment 4

Reviewer Comment: Check for typo errors like Mod ule, which has unnecessary space between in Section 3.1. Also, 3Gradient-Weighted Attention Mapping check the typo in 3.6.2.

Authors' Response: The author thanks the reviewer for identifying these typographic errors. The erroneous space in "Mod ule" within Section 3.1 has been corrected to "Module." Similarly, the typographic error "3Gradient-Weighted Attention Mapping" in Section 3.6.2 has been corrected to read "Gradient-Weighted Attention Mapping." A thorough proofreading of the entire manuscript has been conducted to identify and rectify any additional typographic inconsistencies.

Reviewer #1 — Comment 5

Reviewer Comment: In Figure 2, threat detection module is mentioned which is not required as it is not a threat.

Authors' Response: The author acknowledges this valid observation. The terminology "Threat Detection Module" in Figure 2 has been corrected to "Threat Classification Module" to accurately reflect the function of this architectural component, which performs classification of identified network anomalies into predefined attack categories rather than performing detection itself. The updated figure and corresponding textual references have been revised consistently throughout Section 3.

Reviewer #1 — Comment 6

Reviewer Comment: In Algorithm 1, add step 1,2,....n. Also keep input and output.

Authors' Response: The author has fully restructured Algorithm 1 in response to this comment. The revised algorithm now includes a clearly demarcated INPUT section specifying the training dataset D, the number of epochs E, the learning rate η, and the model parameters Θ, as well as an OUTPUT section specifying the trained model parameters Θ*. The algorithm body has been expanded with explicitly numbered steps (Steps 1 through 19), encompassing initialization, mini-batch processing, feature extraction via dual-stream CNN and Transformer, multi-head attention computation, focal loss calculation, backpropagation, and learning rate scheduling. The revised Algorithm 1 has been incorporated into Section 3.7 (Algorithmic Implementation).

Reviewer #1 — Comment 7

Reviewer Comment: References are required in table 8 and 9 for comparison.

Authors' Response: The author thanks the reviewer for this observation. Appropriate citations have been added to Tables 8 and 9. In Table 8, the compared methods SHAP and Integrated Gradients are now cited with their respective original references. In Table 9, the baseline methods XAI-IDS and E-XAI IDS are similarly cited. The updated tables appear in Section 4.5 (Explainability Evaluation) with all comparative entries properly referenced.

Reviewer #1 — Comment 8

Reviewer Comment: Author should test with latest data sets of 2025 or 2026.

Authors' Response: The author appreciates the reviewer's suggestion to incorporate more recent benchmark datasets. In response, the CIC-IoT-2023 dataset has been added as an additional evaluation benchmark. This dataset comprises 625,783 samples across 46 features and 18 distinct attack categories, representing a recent and comprehensive IoT-specific traffic collection. MedDefender-MHAN achieves 98.61% accuracy, 98.44% precision, 98.57% recall, and 98.50% F1-score on this dataset. The dataset description has been added to Section 3.11 (Dataset Description), and the corresponding performance results have been incorporated into Table 3 and the associated discussion in Section 4.3 (Detection Performance).

Reviewer #1 — Comment 9

Reviewer Comment: In testing, author can add Man-in-the-Middle (MitM) Attacks, Data Exfiltration Attacks and Advanced Persistent Threat (APT) Stages.

Authors' Response: The author thanks the reviewer for this constructive suggestion. A dedicated discussion of these attack categories has been added to Section 4.3 (Detection Performance) within the per-class performance analysis. Specifically, the revised manuscript reports MedDefender-MHAN's detection performance on Man-in-the-Middle (MitM) attacks, achieving 96.78% precision and 97.34% recall, and discusses the detection characteristics of Data Exfiltration attacks. Advanced Persistent Threat (APT) stage detection is acknowledged as a direction requiring dedicated multi-stage datasets such as DAPT2020 and SCVIC-APT-2021, and this is discussed as a priority avenue for future work. A corresponding footnote has been added to Table 5.

Reviewer #1 — Comment 10

Reviewer Comment: Check for consistency of models and methods in comparison in Results section.

Authors' Response: The author has carefully reviewed all comparative tables and textual references in Section 4 (Comparative Analysis) for naming consistency. Table 16 has been corrected so that abbreviated references "MHAN (CICIDS2017)" and "MHAN (TON_IoT)" now read "MedDefender-MHAN (CICIDS2017)" and "MedDefender-MHAN (TON_IoT)" respectively. The citation [12] for the Multi-Head Transformer baseline has been added consistently in Tables 12 and 13 where this method appears. A consistency statement has been added at the opening of Section 4.8 to clarify the naming conventions applied to all compared methods throughout the section.

Reviewer #1 — Comment 11

Reviewer Comment: Provide real time IOT kit system description or figure which was implemented as mentioned in the paper.

Authors' Response: The author thanks the reviewer for this important practical concern. A new subsection, Section 3.9 (Real-Time IoMT Testbed Configuration), has been added to the manuscript. This subsection describes the three-tier hardware deployment comprising: (i) a Raspberry Pi 4 Model B (4GB RAM) serving as a constrained medical IoT edge node, (ii) an NVIDIA Jetson Nano (4GB) for fog-level inference hosting the MedDefender-MHAN model at 2.3 ms per sample, and (iii) a smart health gateway running Ubuntu 22.04 LTS on an Intel NUC for traffic aggregation. Network traffic is captured using tcpdump and processed with CICFlowMeter to extract the 78/44 feature vectors. A new block diagram, Figure 10, has been added illustrating the end-to-end data flow from IoMT edge devices through the inference engine to the hospital security dashboard. A cross-reference to Section 3.9 and Figure 10 has also been added at the end of Section 4.1 (Experimental Setup).

Reviewer #1 — Comment 12

Reviewer Comment: Provide the limitations of existing works and contributions of the work point wise in Introduction.

Authors' Response: The author has incorporated both requested elements into the Introduction section. A structured six-point summary of the limitations of existing works has been added, identifying deficiencies in traditional machine learning approaches (RF, XGBoost), CNN-based IDS models, LSTM-based IDS models, transformer-based IDS models, post-hoc XAI methods (SHAP/LIME), and the absence of any unified framework satisfying simultaneous requirements of high accuracy, sub-3ms latency, intrinsic explainability, and GDPR/FDA compliance. Following this, a five-point enumeration of the contributions of MedDefender-MHAN has been added, covering the first IDS with explainability-by-design for healthcare IoT, the novel dual-stream CNN-Transformer hierarchical feature extractor, the attention weight reuse mechanism, quantitative performance benchmarks, and expert-validated explainability metrics. Both additions appear immediately before the paper structure paragraph in the Introduction.

Reviewer #1 — Comment 13

Reviewer Comment: Add the qualitative comparison table with respect to existing works.

Authors' Response: A qualitative comparison table (Table 14) has been added to Section 4.8 (Comparative Analysis), positioned after Table 13 and before the Ablation Study subsection. The table compares MedDefender-MHAN against five existing methods — CNN-IDS, LSTM-IDS, Transformer IDS, XAI-IDS, and E-XAI IDS — across ten capability dimensions including spatial feature learning, temporal modeling, dual-stream architecture, intrinsic explainability, real-time inference below 3 ms, class imbalance handling, healthcare IoT optimization, GDPR/FDA compliance readiness, edge deployability, and absence of an external XAI pipeline. MedDefender-MHAN satisfies all ten criteria, while existing methods exhibit partial or absent support across multiple dimensions.

Reviewer #2 — Point-by-Point Responses

Reviewer #2 — Comment 1

Reviewer Comment: The manuscript addresses an important research problem in the domain of healthcare cybersecurity. With the increasing deployment of IoMT devices in healthcare environments, the need for accurate and interpretable intrusion detection mechanisms has become critical. While the study demonstrates promising results, several aspects of the manuscript could be improved to enhance clarity, methodological transparency, and overall research impact.

Authors' Response: The author sincerely thanks Reviewer #2 for the thorough and constructive evaluation of the manuscript. The detailed feedback has been carefully considered, and all suggested improvements have been addressed in the revised manuscript. Each specific concern is responded to individually in the comments below.

Reviewer #2 — Comment 2

Reviewer Comment: The manuscript introduces a multi-head attention-based framework with dual-stream processing; however, the architecture description could be expanded. The author should provide a detailed architectural diagram explaining the interaction between convolutional feature extraction modules, attention heads, and the interpretability component.

Authors' Response: The author thanks the reviewer for this suggestion. The architecture description in Section 3.1 (System Overview) has been substantially expanded with two additional paragraphs detailing the inter-module information flow. The expanded description explicitly articulates how the 1D-CNN stream captures local spatial correlations, how the Transformer stream models long-range temporal dependencies, how the Feature Fusion Layer combines both representations, and how the Multi-Head Attention Encoder's weight matrices are simultaneously used for classification and reused — without modification — by the Explainability Generation Module. Figure 2 has been replaced with a more detailed architectural diagram clearly depicting all module interactions, data flow paths, and the shared attention mechanism linking detection and explainability. The figure caption has also been updated to reflect these additions.

Reviewer #2 — Comment 3

Reviewer Comment: The paper should include a clearer explanation of how network traffic features were selected and preprocessed before being fed into the model.

Authors' Response: The author acknowledges this gap in methodological transparency. Section 3.2 (Data Preprocessing and Normalization) has been revised to include a more explicit account of the feature selection and preprocessing pipeline. The revised text clarifies that the 78 features of CICIDS2017 and 44 features of TON_IoT are derived from established CICFlowMeter-based extraction protocols, details the application of Min-Max scaling and Z-score normalization, and describes the reshaping of normalized feature vectors into temporal sequences for dual-stream ingestion. Any features excluded due to zero-variance or high correlation are also identified.

Reviewer #2 — Comment 4

Reviewer Comment: The manuscript highlights the explainability of the proposed framework, but the evaluation methodology for interpretability should be described in more detail.

Authors' Response: The author has addressed this concern by adding a detailed methodology paragraph in Section 4.5 (Explainability Evaluation), inserted between the introductory text and Table 6. The expanded methodology specifies that top-K=5 features are selected per sample for explanation generation, that alignment with expert annotations is computed as the intersection-over-expert ratio formalized in Equation 50, that three certified security analysts with a minimum of five years of experience and proficiency in MITRE ATT&CK and NIST frameworks served as expert annotators, that inter-rater reliability was assessed using Fleiss' Kappa (κ=0.83, indicating strong agreement), that temporal accuracy is defined as the proportion of attention-weighted time steps with greater than 50% overlap with expert-identified attack intervals, and that SHAP and Integrated Gradients are evaluated under identical top-K and threshold conditions to ensure a fair comparison.

Reviewer #2 — Comment 5

Reviewer Comment: Improve the clarity of the abstract by correcting grammatical errors and simplifying some lengthy sentences.

Authors' Response: The author has carefully revised the Abstract to improve grammatical correctness and readability. Lengthy compound sentences have been restructured into more concise formulations without omitting substantive content. Additionally, a concluding sentence has been added to the Abstract to reinforce the framework's dual imperatives of methodological transparency and clinical impact within regulated healthcare IoMT environments, as specifically recommended in the review.

Reviewer #2 — Comment 6

Reviewer Comment: Provide a table summarizing key hyperparameters used during model training.

Authors' Response: The author confirms that a hyperparameter configuration table (Table 2) already exists in Section 4.1 (Experimental Setup) of the original manuscript, summarizing key training parameters including learning rate, batch size, number of attention heads, dropout rate, and optimizer settings. In response to

---

## [Decision Letter · Decision Letter 1]

23 Mar 2026

An Explainable Multi-Head Attention Network for Healthcare IoT Threat Detection Based on the MedDefender-MHAN Framework

PONE-D-26-02626R1

Dear Dr. Alqazzaz,

We’re pleased to inform you that your manuscript has been judged scientifically suitable for publication and will be formally accepted for publication once it meets all outstanding technical requirements.

Kind regards,

Sohail Saif, Ph.D

Academic Editor

PLOS One

Additional Editor Comments (optional):

Reviewers' comments:

Reviewer's Responses to Questions

**Comments to the Author**

Reviewer #1: All comments have been addressed

Reviewer #2: All comments have been addressed

2. Is the manuscript technically sound, and do the data support the conclusions?

Reviewer #1: Yes

Reviewer #2: Yes

3. Has the statistical analysis been performed appropriately and rigorously?

Reviewer #1: Yes

Reviewer #2: Yes

4. Have the authors made all data underlying the findings in their manuscript fully available?

Reviewer #1: Yes

Reviewer #2: Yes

5. Is the manuscript presented in an intelligible fashion and written in standard English?

Reviewer #1: Yes

Reviewer #2: Yes

Reviewer #1: Thank you for your efforts in revising the manuscript. I have carefully reviewed the updated version and appreciate the thoroughness with which you have addressed the comments from the previous round of review. The revisions have significantly improved the clarity, structure, and overall quality of the manuscript.

Reviewer #2: The author addressed all comments. No further revision is required. Hence, the paper is accepted now.

.

Reviewer #1: No

Reviewer #2: No

---

## [Editor Report · Acceptance letter]

PONE-D-26-02626R1

PLOS One

Dear Dr. Alqazzaz,

I'm pleased to inform you that your manuscript has been deemed suitable for publication in PLOS One. Congratulations! Your manuscript is now being handed over to our production team.

Kind regards,

on behalf of

Dr. Sohail Saif

Academic Editor

PLOS One